# Large Language Model-based Data Science Agent: A Survey

**Ke Chen**[†]                                    *kec10@illinois.edu*
*University of Illinois Urbana-Champaign*

**Peiran Wang**[†]                                *whilebug@gmail.com*
*University of Illinois Urbana-Champaign*

**Yaoning Yu**[†]                                 *yyn003600@gmail.com*
*University of Illinois Urbana-Champaign*

**Xianyang Zhan**                                 *zhan39@illinois.edu*
*University of Illinois Urbana-Champaign*

**Haohan Wang**                                   *haohanw@illinois.edu*
*University of Illinois Urbana-Champaign*

**Reviewed on OpenReview:** *https://openreview.net/forum?id=ZT5SJQN0CS*

## Abstract

The rapid advancement of Large Language Models (LLMs) has driven novel applications across diverse domains, with LLM-based agents emerging as a crucial area of exploration. This survey presents a comprehensive analysis of LLM-based agents designed for data science tasks, summarizing insights from recent studies. From the agent perspective, we discuss the key design principles, covering agent roles, execution, knowledge, and reflection methods. From the data science perspective, we identify key processes for LLM-based agents, including data preprocessing, model development, evaluation, visualization, etc. Our work offers two key contributions: (1) a comprehensive review of recent developments in applying LLM-based agents to data science tasks; (2) a dual-perspective framework that connects general agent design principles with the practical workflows in data science.

## 1 Introduction

In recent years, the rapid development of Large Language Models (LLMs) has driven significant innovations across various domains. Leveraging their remarkable capabilities in understanding and generating human-like text, LLMs have become foundational in creating intelligent agents capable of performing complex tasks autonomously. These agents have demonstrated substantial potential in diverse fields, including healthcare Qiu et al. (2024), finance Yu et al. (2024), education Zhang et al. (2025b), and software engineering Hong et al. (2023).

Among these fields, data science has emerged as a particularly critical area for applying LLM-based agents Sun et al. (2024b). Data science involves extracting meaningful insights from vast and diverse datasets, a process that traditionally requires extensive manual effort and expertise. Consequently, LLM-based data science agents (DS Agents) have attracted attention for their ability to automate and optimize data analysis, model development, and decision-making processes.

A central goal of this survey is to clarify how LLM agents can be organized to perform the functions of a data scientist. This leads to a sequence of increasingly detailed research questions. (RQ1) At the organizational level, how should agents' roles and responsibilities be defined so that complex analytical tasks

---

[1]† These authors contributed equally to this work.

can be effectively decomposed? (RQ2) Given these roles, how should each agent carry out its reasoning and tool use to ensure reliable execution? (RQ3) Execution usually depends on information beyond the agent's internal context. What external knowledge should agents access, and how should such knowledge be integrated into their decisions? (RQ4) Even with access to the necessary knowledge, agents still make mistakes, raising the question of what reflection and self-correction mechanisms are required to detect and repair errors during operation. (RQ5) Finally, data science workflows are inherently iterative, prompting the question of how these agent-level capabilities can support end-to-end, loop-based processes rather than isolated single-step tasks.

To address these research questions, we examine LLM-based data science agents from two complementary perspectives: agent design and data science application. From the agent design perspective, we summarize key architectural paradigms—including single-agent systems, collaborative multi-agent structures, and dynamic agent generation—and analyze core components relevant to RQ1–RQ4, including how the agent structure is designed (§3.1), how the reasoning of LLM within agents is performed (§3.2), where the knowledge comes from (§3.3), and the reflection of the agent (§3.4). such as agent roles, execution strategies, knowledge integration, and reflection mechanisms. From the data science perspective(§4), we explore how LLM agents are applied across major workflow stages such as data preprocessing, modeling, evaluation, and visualization. We also outline common task types, including model-building and insight-generation tasks, and characterize the iterative nature of the data science loop. This perspective directly informs RQ5 by illustrating how LLM agents support each stage of an end to end data science workflow. Additionally, our survey goes beyond mere documentation by synthesizing insights from recent studies to identify research opportunities and future directions in this evolving field Sahu et al. (2024); Fan et al. (2023).

In summary, this survey makes two primary contributions.

- It provides a comprehensive review of recent efforts to apply LLM-based agents to data science tasks, synthesizing work across areas such as data preprocessing, modeling, evaluation, and visualization.

- It proposes a dual-perspective framework that bridges the gap between general agent design principles—such as role allocation, execution, and reflection—and the specific operational needs of data science workflows, offering a structured lens to understand and develop LLM-based data science systems.

## 2 Related Works

Recent surveys on LLM-based multi-agent systems have introduced various taxonomies from architectural, task-specific, and coordination perspectives. We briefly review them and highlight how our approach differs.

### 2.1 Modular Architectures

Existing surveys often decompose LLM-based agents into functional modules such as planning, memory, perception, and action. For example, Guo et al. (2024b) identifies components like agent-environment interface and capability acquisition; similar structures appear in Liu et al. (2024a), Sun et al. (2024b), and others.

Our work adopts this modular view but anchors it in the context of data science workflows. Instead of listing capabilities, we emphasize how modules interact during task execution—e.g., how reasoning, planning, and knowledge access are coordinated in dynamic or static execution (§3.2), and how external knowledge sources are integrated into decision-making (§3.3). We also highlight how modularity supports runtime adaptation through reflection mechanisms(§3.4), where agents adjust behaviors based on feedback, errors, or performance signals—enabling dynamic coordination across modules.

### 2.2 Collaboration and Communication

Agent collaboration is commonly categorized by structure (centralized, decentralized) or mode (cooperation, competition, etc.), as seen in Tran et al. (2025). Related works Guo et al. (2024b), Li et al. (2024g) discuss role-based or layered designs.

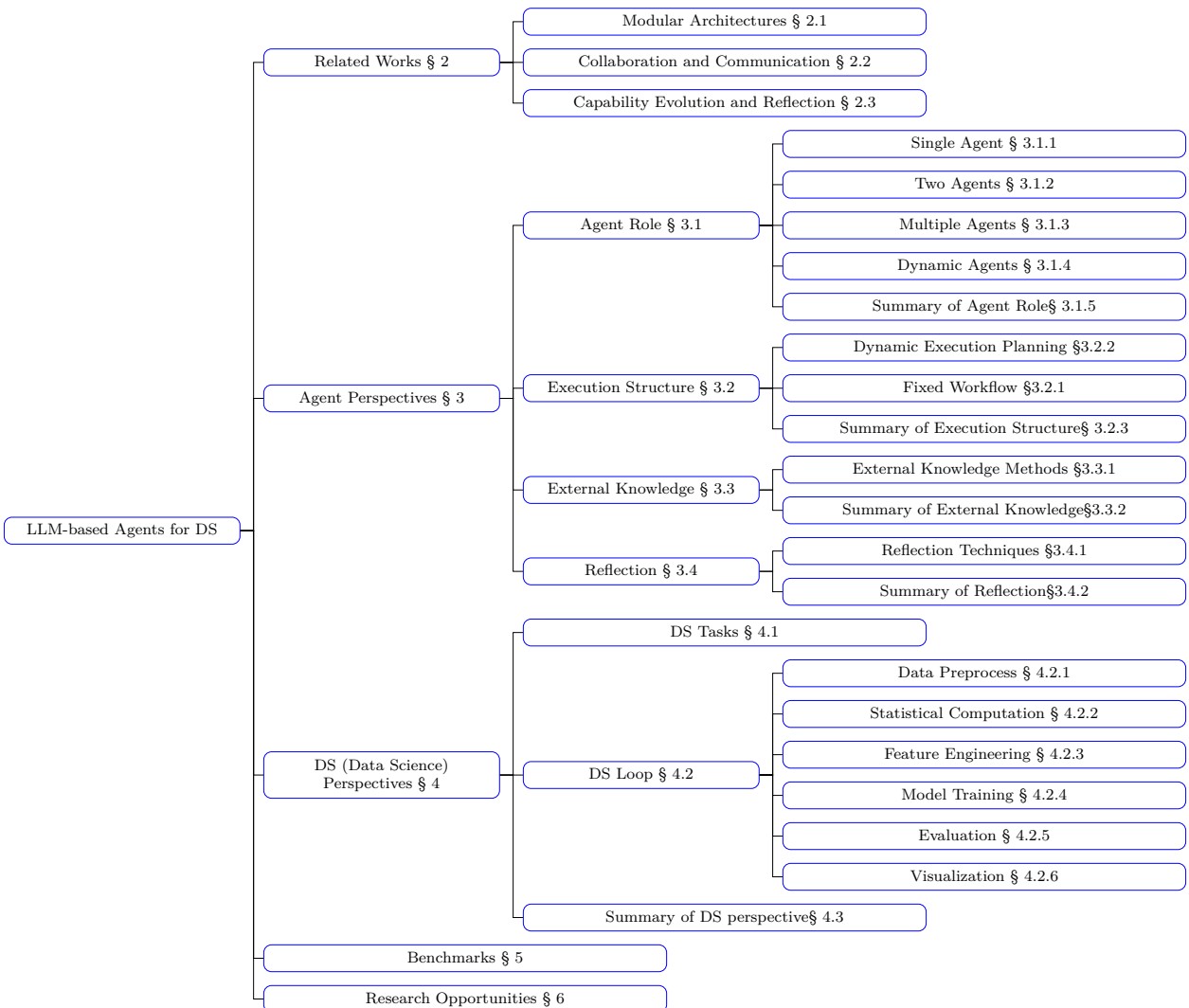

Figure 1: Structure of This Survey

However, these are mostly static views. In our work, we also examine dynamic orchestration, where agents adaptively coordinate by adjusting roles or workflows during execution. This includes settings where task allocation evolves based on feedback, or agents are restructured at runtime to respond to changing demands (§3.1.4, §3.2).

## 2.3 Capability Evolution and Reflection

Several surveys recognize that agent systems can adapt via feedback, memory updates, or reflection—for example, Li et al. (2024g) includes an evolution phase with memory consolidation, and Wang et al. (2024a) discusses reflective planning. Similar notions appear in Guo et al. (2024b) and others.

In contrast to treating reflection as an auxiliary feature, we emphasize it as a cross-stage mechanism that drives dynamic execution adjustments (§3.4). We outline three key dimensions: the driver, level(scope of impact), and adaptability of reflection, framing reflection as a central control process for progress monitoring and adaptive behavior.

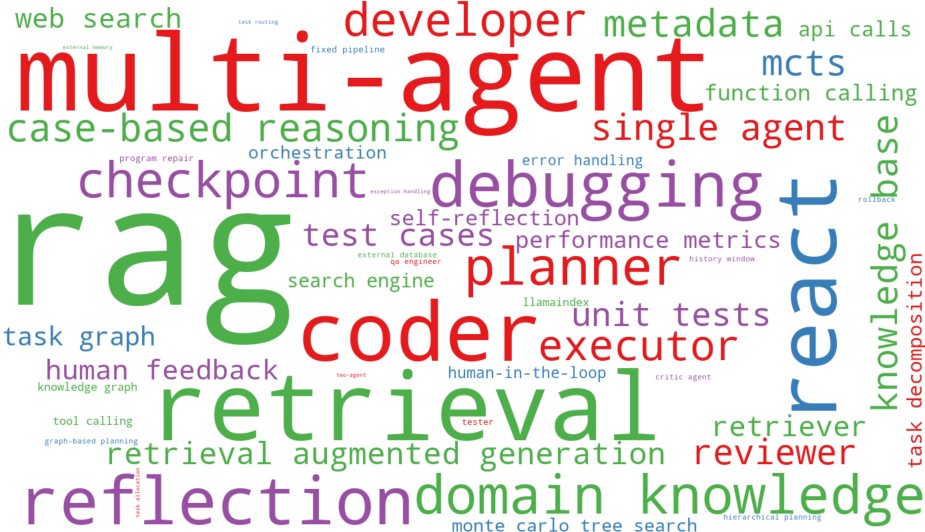

Figure 2: Keyword cloud summarizing the most frequent concepts across the surveyed LLM-based data science agent papers. Larger words represent higher occurrence frequency. The colors in the cloud correspond to the major analysis dimensions discussed in this survey: red words represent agent role design (§3.1), blue words represent execution structure (§3.2), green words represent external knowledge integration (§3.3), and purple words represent reflection mechanisms (§3.4).

# 3 Analysis from Agent Perspective

LLMs provide strong linguistic and generative capabilities, but they exhibit structural limitations that become particularly evident in data science workflows. First, LLMs are prone to hallucination, e.g., fabricating column names, statistical results, or entire processing steps. Second, LLM-generated code is brittle. Even when logically correct in natural language, produced scripts frequently break due to issues such as type inconsistencies, missing dependencies, unstable pipelines, or non-reproducible execution paths. Third, LLMs struggle with long-horizon tasks. Their limited ability to retain and manipulate extended context leads to goal drift, forgotten assumptions, and inconsistencies across multi-step analytical processes. These limitations show that raw LLMs are not reliable as end-to-end data science systems and need structured agent architectures to provide the capabilities they lack and to address common failure modes.

Motivated by these limitations, recent work has increasingly turned to agent architectures that introduce structure, modularity, and external support, enabling LLMs to compensate for their inherent limitations. For example, grounding and verification mechanisms help reduce hallucination, controlled execution improves reliability, and explicit role decomposition mitigates long-context instability. Agent systems thus do not merely wrap LLMs—they provide the organizational scaffolding needed to transform a general-purpose model into a dependable data science practitioner.

Seen from this perspective, our analysis focuses on four design dimensions: agent role design (§3.1), execution structure (§3.2), external knowledge integration (§3.3), and reflection mechanisms (§3.4). They form a structured solution to the inherent limitations of LLMs in data science settings. The remainder of this section examines each dimension in turn, showing how existing systems operationalize these design principles in practice.

## 3.1 Agent Role Design

In this section, we discuss the role design of LLM-based agents, focusing on their agent role specification. Starting from single-agent designs, which manage all tasks independently, we summarize the transition to two-agent systems that introduce role separation, such as planner-executor and coder-reviewer frameworks.

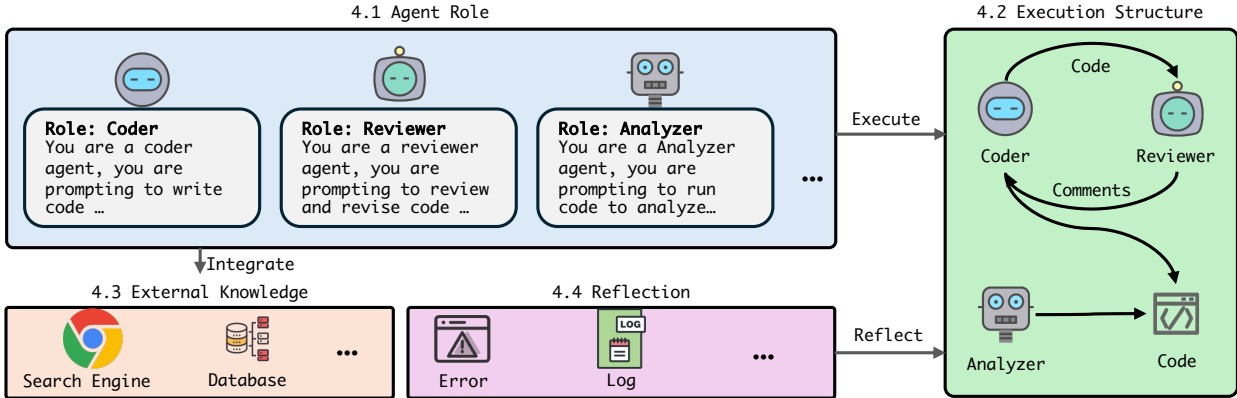

Figure 3: We illustrate the basic components for current data science agents: 1) agent role; 2) execution structure; 3) knowledge and 4) reflection.

Furthermore, we conclude different multi-agent systems, covering software engineering style systems, minimum function agents, etc. Lastly, we introduce some dynamic agent role design frameworks where agents are generated adaptively rather than predefined.

### 3.1.1 Single Agent

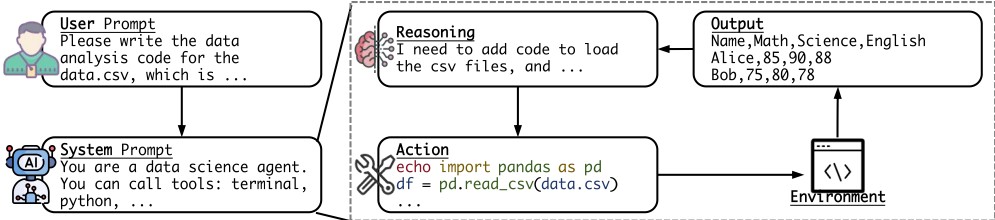

Figure 4: The basic structure for a single agent structure, with only the agent and execution environment.

The single-agent design is the simplest design structure as shown in Figure 4. Most existing single-agent works directly follow the ReAct design (a prompting technique , where LLMs are used to generate both reasoning traces and task-specific actions) Freimanis & Andersson Rhodin (2024); Chen et al. (2024a); Sun et al. (2024a); Jing et al. (2024); Hu et al. (2024a); Deng et al. (2024); Le et al. (2023); Zhang et al. (2024a); Gupta et al. (2024)Bendinelli et al. (2025); Xu et al. (2025), where the single agent will perform thought, action, and observation processes iteratively on its own.

Single-agent designs offer minimal overhead and are easy to deploy, making them attractive for low-latency or low-cost settings. However, since all reasoning, memory, and execution are concentrated in a single model, they inherit most of the LLM limitations: hallucinations are unmitigated, generated code is fragile, and long-horizon tasks often suffer from goal drift and lost context. As a result, single-agent systems are often insufficient for complex or multi-stage workflows that require long-horizon reasoning, consistent state tracking, or robust code execution.

### 3.1.2 Two Agents

Evolving from the single agent, the two-agent design is introduced to decompose the single agent's functions, dividing the functions into two specialized roles, particularly either a planner&executor or code&reviewer structure.

**Planner and executor**. Different from the single agent discussed previously, some works split the single agent into two agents Huang et al. (2024b); Liu et al. (2024b); Zhang et al. (2023b); Chi et al. (2024); Li et al. (2024e)Wang et al. (2025a), planner and executor. As an example shown in Figure 5, the planner will get observations from previous execution results or users' requests, and generate a next-step plan, or a

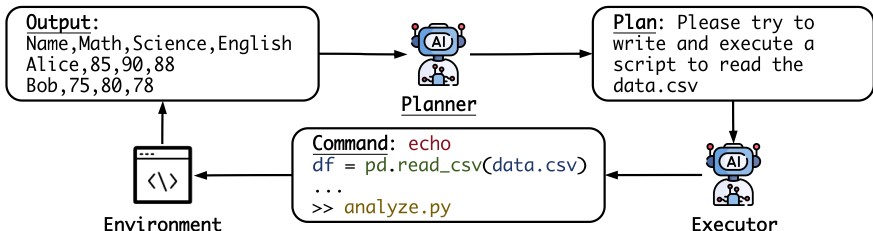

Figure 5: An example of planner and executor agent structure, where the planner generates a plan for the executor to execute in detail.

whole plan in advance. Then, the executor is prompted to interact with the environments and will follow the generated plans to execute.

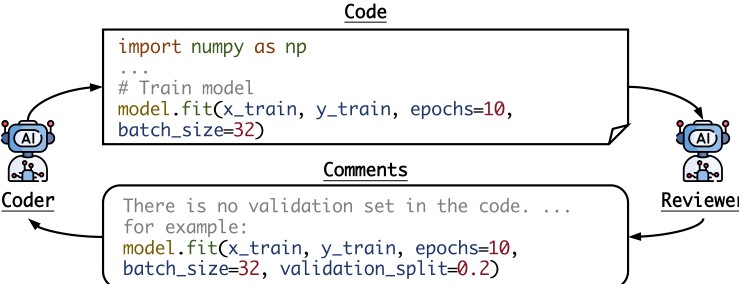

Figure 6: In the coder&reviewer structure, the coder generates the code, while the reviewer will make comments to revise the code for the coder.

**Coder and reviewer**. Another type of two-agent structure is the "coder and reviewer" style Trirat et al. (2024); Huang et al. (2023) as shown in Figure 6. In such a design, the coder is responsible for completing tasks following the typical ReAct structure. Another agent, the reviewer is introduced to check the validity of the code generated by the coder. Trirat et al. (2024) allows the reviewer to check the generated response at each step of generation, while Huang et al. (2023) only allows the reviewer to function at the end of the entire generation process.

Two-agent designs improve over single-agent systems by splitting responsibilities into complementary roles. Planner–executor structures provide more structured task decomposition, and coder–reviewer structures introduce lightweight verification, catching some errors or hallucinations that a single agent would miss. However, the approach remains sensitive to failures such as a planner hallucinating an incorrect plan or a reviewer failing to identify subtle logical errors. Communication cost also increases, and the reliability gains depend heavily on how well the two roles align. As a result, two-agent systems offer a moderate balance between simplicity and robustness, but their effectiveness is bounded by the underlying LLM's consistency and self-correction ability.

### 3.1.3 Multiple Agents

Multi-agent systems enhance problem-solving capabilities by enabling collaboration among multiple agents, each with distinct roles and expertise. In such systems, agents are assigned specialized responsibilities, allowing them to focus on different tasks while exchanging progress and information.

**Software engineering-style team**. The Software Engineering (SE) team-style agent design draws inspiration from traditional human software development teams Qian et al. (2024); Zhao et al. (2024b); Trirat et al. (2024); Hong et al. (2023); Tao et al. (2024); Lin et al. (2024); Nguyen et al. (2024). In an SE team-style framework, agents are typically assigned roles that correspond to key human roles in software development, here we state some example roles: The product manager defines the product vision and organizes requirements. The requirements analyst translates user needs into detailed software specifications. The scrum master facilitates task planning and sprint coordination. Hierarchical roles like team leader, module Leader,

and function coordinator handle task decomposition at varying levels of granularity. The executor for the task is the developer, who is responsible for implementing the code, while the senior developer refines it. Finally, the QA engineer ensures quality through rigorous testing, and the tester validates specific functionalities. Some SE team-style agents also perform hierarchical roles which allow some agents with high permissions to manage others.

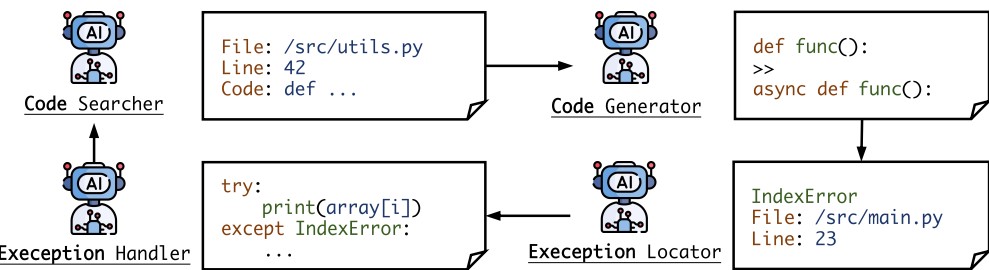

Figure 7: For agents with minimum functions, each agent is only responsible for a minimum function, such as search code, run code, etc.

**Minimum function agents**. Minimum function agents are designed to handle narrowly scoped and atomic tasks separately, as shown in Figure 7.

Across existing frameworks, minimum function agents are given very small and specific functions to handle. For instance, code search agents locate relevant files, classes, or methods within a repository, while fault localization agents identify buggy code sections using debugging techniques like spectrum-based fault localization Zhang et al. (2024e). Other agents specialize in generating patches to fix identified issues, executing tests to validate code correctness, or systematically building repository structures from high-level descriptions Zan et al. (2024); Arora et al. (2024). In the context of exception handling, some agents detect fragile code, identify exception types, and implement robust handling mechanisms to enhance code reliability Zhang et al. (2024b). All works emphasize task decomposition and modularity, with outputs from one agent often serving as inputs for another, forming structured and collaborative workflows Zan et al. (2024); Phan et al. (2024); Arora et al. (2024)Seo et al. (2025); You et al. (2025); Ou et al. (2025).

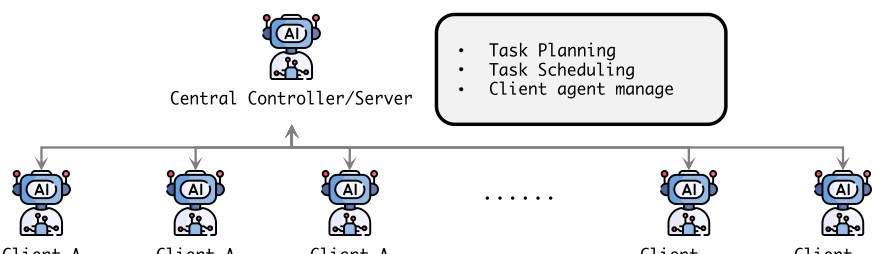

Figure 8: In the client-server agent structure, normally there will be a central server controls all the other clients.

**Client-server design**. Client-server agents, adopt a hierarchical architecture where a central controller agent manages and coordinates the operations of multiple specialized client agents Zhang et al. (2023a); Yang et al. (2024a); Gandhi et al. (2024); Zhao et al. (2024a).

The controller or server agent functions as a project manager, planning entire workflows of the given tasks, and allocates the tasks to the client agents. Client agents perform as roles like software engineers or testers, focusing on executing specific subtasks such as data analysis, modeling, or feedback generation. These roles allow for clear separation of tasks, while also enabling dynamic adjustments based on task requirements or external feedback Zhang et al. (2023a); Bai et al. (2024); Zhao et al. (2024a); Shen et al. (2024).

Multi-agent architectures offer the highest scalability and strongest resistance to long-context forgetting, as each agent operates within a narrowly scoped role. They can achieve high reliability through division of labor, modular workflows, and cross-agent verification. Yet the increased number of agents introduces

coordination overhead, susceptibility to miscommunication, and longer execution latency. These systems are powerful but harder to control, and their effectiveness depends on careful manual workflow design.

### 3.1.4 Dynamic Agents

Dynamic agents represent a class of multi-agent systems that emphasize adaptability by dynamically creating, modifying, or expanding agents during runtime. Unlike static agents, which follow predefined workflows and configurations, dynamic agents are designed to respond to the complexity or variability of tasks by adjusting the agents' internal structure (prompt, etc.) or introducing new agents Hu et al. (2024b); Ishibashi & Nishimura (2024). Current frameworks for dynamic agent creation adopt two primary paradigms:

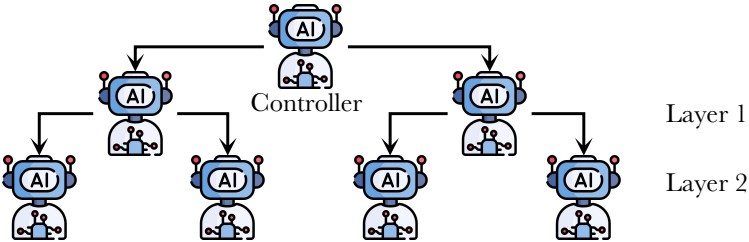

Figure 9: Hierarchical agent generation.

**Hierarchical agent generation**. This paradigm involves a parent agent or a high-level controller (e.g., Mother Agent in SoA Ishibashi & Nishimura (2024)) that decomposes complex tasks into subtasks and creates child agents to handle the subtasks. Each child agent operates independently on its specific subtask. It is particularly effective in managing tasks with clear functional divisions, such as modular code generation or system-level software design Ishibashi & Nishimura (2024).

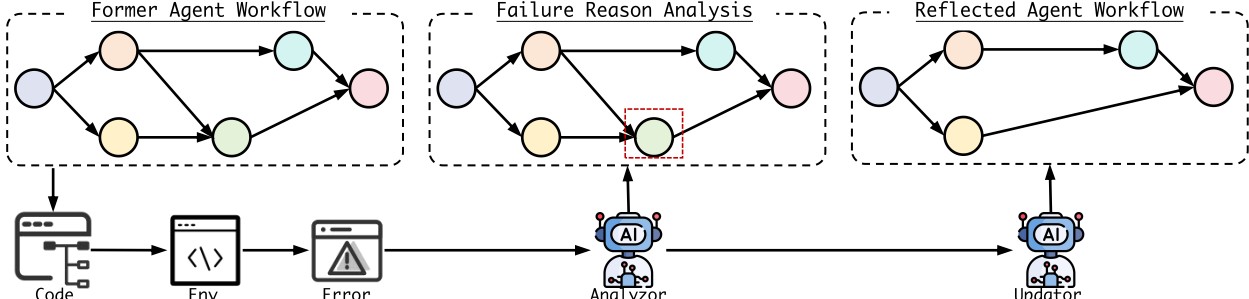

Figure 10: Iterative agent generation through feedback.

**Iterative agent generation through feedback**. Dynamic agents adjust their structures and behaviors iteratively based on real-time feedback from their environment or other agents (Figure 10). EvoMAC Hu et al. (2024b) exemplifies this paradigm, employing a collaborative rather than hierarchical approach, where agents refine their outputs and workflows through mechanisms analogous to backpropagation—such as a loss agent computing errors and an update agent adjusting agent workflows accordingly. This textual feedback enables the dynamic creation or reconfiguration of agents, supporting adaptability for tasks with evolving requirements, such as changing software specifications (Figure 10) Hu et al. (2024b).

Dynamic agent generation offers high flexibility by creating agents on demand rather than relying on manually designed roles, prompts, and pipeline. This allows the system to adjust its structure at runtime and better accommodate diverse or evolving data science tasks. However, dynamically spawning agents also amplifies LLM uncertainty, reduces reproducibility, and increases computational and coordination overhead. As a result, these systems provide strong adaptability but exhibit lower predictability and stability, making them challenging to deploy in stable production settings.

### 3.1.5 Summary of Agent Role

Beyond specific agent role design, we summarize the key features of agent roles, including agent structure, agent relationship, role task allocation, and task granularity. These key features reveal the core thoughts through the design of agent roles.

| Framework | CS | FL | PG | VA | RI | ED | EH |
|---|---|---|---|---|---|---|---|
| AutoCodeRover | ✓ | ✓ | ✓ | ✓ | | | |
| CODES | | | | | ✓ | | |
| HYPERAGENT | ✓ | | | ✓ | | | |
| MASAI | ✓ | ✓ | ✓ | ✓ | | | |
| Seeker | | | | | | ✓ | ✓ |

Table 1: This table summarizes the roles of Minimum Function Agents in different frameworks. The columns represent specific functions: Code Search (CS), Fault Localization (FL), Patch Generation (PG), Validation (VA), Repository Initialization (RI), Exception Detection (ED), and Exception Handling (EH). A checkmark (✓) indicates that the framework supports the corresponding function.

| Role | PM | RA | AR | SM | TL | ML | FC | DE | SD | QA | TE |
|---|---|---|---|---|---|---|---|---|---|---|---|
| AgileCoder | ✓ | | | ✓ | | | | ✓ | ✓ | | ✓ |
| AutoML-Agent | ✓ | | | | | | | | | ✓ | |
| ChatDev | | ✓ | | | | | | ✓ | | | ✓ |
| FlowGen | | ✓ | ✓ | ✓ | | | | ✓ | | | ✓ |
| MetaGPT | ✓ | | ✓ | | | | | ✓ | | ✓ | ✓ |
| MAGIS | ✓ | | | | | | | ✓ | | ✓ | |
| VisionCoder | | | | | ✓ | ✓ | ✓ | ✓ | | | ✓ |

Table 2: Summary of SE Team Roles in Agent Designs: Product Manager (PM), Requirements Analyst (RA), Architect (AR), Scrum Master (SM), Team Leader (TL), Module Leader (ML), Function Coordinator (FC), Developer (DE), Senior Developer (SD), QA Engineer (QA), Tester (TE).

**Agent structure**. The structure of LLM-based agents can be categorized into several types, each with its advantages and challenges:

1. With a manager: In this structure, a central agent manages and controls all agents. Software engineering-style agents and Client-server agents mainly have this structure. For example, in the AutoML-GPTTrirat et al. (2024) framework, a central LLM serves as the controller, managing the entire pipeline by integrating specialized agents for subtasks such as model design and hyperparameter tuning.

2. Without a manager: Each agent operates independently and solves tasks autonomously. Minimum function agents mainly pose this structure, since all the agents with minimum function share the same position. An example of this is the MASAI Arora et al. (2024) framework, which utilizes decentralized agents that collaborate on machine learning and data science tasks by sharing results but not a central management system.

3. Hierarchical managers: A higher-level agent controls lower-level agents in a layered structure. Hierarchically generated agents and part of software engineering style agents mainly pose such structure. An example of this can be found in Hierarchical agent generation, such as in EvoMAC Hu et al. (2024b), where a parent agent dynamically creates child agents to handle specific subtasks during runtime.

**Agent relationship**. The relationship between agents within the system can vary significantly depending on the design philosophy:

1. Compete: Agents work against each other to complete a task, often in the coder-reviewer paradigm (similar to the adversarial concept in GAN). In frameworks like MASAI Arora et al. (2024), agents engage in competitive strategies to address machine learning and data science challenges. The competition

| Dimension | Single Agent | Two-Agent | Multi-Agent | Dynamic Agents |
|---|---|---|---|---|
| Reliability | ○ | ◐ | ● | ◐ |
| Scalability | ○ | ◐ | ● | ● |
| Coordination Cost | ● | ◐ | ○ | ○ |
| Predictability / Stability | ◐ | ◐ | ● | ○ |
| Industrial Applicability | Quick analyses, ad-hoc queries, and low-risk automation | Medium-scale workflows requiring basic verification and predictable execution | Production pipelines with modular stages, quality checks, and stable data dependencies | Exploratory analytics, quick pipeline prototyping, or changeable environment |

Table 3: Trade-off summary across agent role designs. ●= strong, ◐= moderate, ○= weak.

between reviewers and coders allows the iterative refinement of the code. Some works also allow multiple agents to propose multiple plans to compete.

2. Collaborate: Agents work together toward a shared objective. For example, in the MAGIS Tao et al. (2024) framework, agents assume different roles like Manager, Developer, and QA Engineer to collaborate on resolving GitHub issues. Their tasks are divided to ensure modular development, with continuous collaboration between agents.

3. Hybrid:

   Agents alternate between competing and collaborating based on the task requirements. For instance, in AutoCodeRover Zhang et al. (2024e), agents work together to localize faults and generate patches, but may compete in terms of optimizing solutions or strategies based on the specific issue at hand.

**Agent role task allocation**. LLM-based agents can allocate tasks in either a static or dynamic manner:

1. Static Task Allocation: In some systems, agents are assigned a fixed set of tasks that they perform. For example, in the Data Director Hong et al. (2024) framework, agents follow a static task allocation, where the tasks are predefined and agents work through structured stepwise execution.

2. Dynamic Task Allocation: Tasks are allocated based on the real-time needs and feedback from the system or environment. An example of dynamic task allocation can be seen in the EvoMAC Hu et al. (2024b) framework, where agents adjust dynamically based on environmental feedback, creating or dismissing agents as needed to refine or expand their tasks.

**Agent Role Task Granularity**. The granularity of tasks assigned to agents influences both the precision and complexity of their execution:

1. Coarse Granularity: Some agents are given broader, less detailed tasks. For example, in AutoML-GPT Trirat et al. (2024), the central controller agent coordinates the entire machine learning pipeline, handling coarse-grained tasks such as managing the overall workflow rather than focusing on the details of each individual task.

2. Fine Granularity: In other cases, tasks are broken down into smaller units for more specific execution. For example, in MapCoder Islam et al. (2024a), multiple agents collaborate, each handling a fine-grained task such as code generation, debugging, or retrieval. This detailed task assignment ensures high accuracy but requires more computation overhead.

## 3.2 Execution Structure

In this section, we summarize the execution strategies of LLM-based agents, emphasizing their approaches how to complete the tasks.

### 3.2.1 Static Execution

Static execution refers to a workflow-style structure where agents follow a predefined sequence of actions to accomplish tasks. In this paradigm, the workflow is rigidly designed, ensuring that each step is executed deterministically without deviations. This type of workflow is particularly useful in scenarios where tasks require consistent, repeatable processes or involve complex subtasks that must be systematically handled. By defining clear workflows, these systems ensure reliability, transparency, and ease of evaluation, as agents operate within well-defined boundaries. A common feature of static execution structures is the division of tasks into sequential, predefined stagesSeo et al. (2025); Li et al. (2025); Xu et al. (2025); Ou et al. (2025). These workflows often start with data or task interpretation, followed by intermediate processing steps such as feature selection, data transformationQi & Wang (2024), or subtask decomposition Luo et al. (2024), and conclude with result generation Gu et al. (2024) or validation Shen et al. (2024).

### 3.2.2 Dynamic Execution

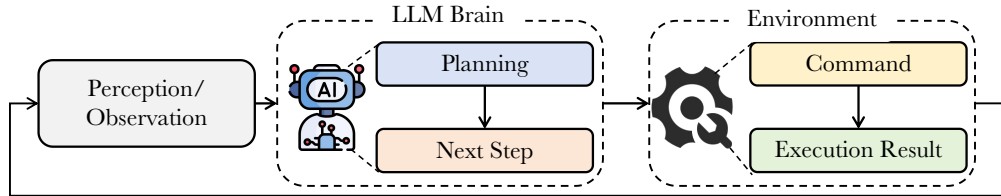

Figure 11: In just-in-time planning, one agent is responsible for planning and execution simultaneously.

**Just-in-time plan**. The "just-in-time" planning approach represents a dynamic and iterative execution strategy widely adopted by modern LLM-based agents (see Figure 11). Unlike pre-defined static workflows, this structure enables agents to generate and refine plans based on real-time feedback from previous execution steps Rasheed et al. (2024); Yang et al. (2024a)Bendinelli et al. (2025). Specifically, agents observe the outcomes of each executed step, use these observations to reevaluate the task's context, and then dynamically devise the next step Cao et al. (2024); Zhao et al. (2024a). Just-in-time planning is particularly effective in domains where the environment or task requirements are dynamic Zhao et al. (2024a), as it reduces redundancy and enhances precision by continuously aligning actions with the latest data or results Zhang et al. (2024d); Liu et al. (2024b).

**Plan then execute**. The "plan-then-execute" framework is a widely adopted structure in LLM agent systems, particularly for tasks requiring complex reasoning and multi-stage problem-solving (see Figure 5 for an example). This framework divides the agent's workflow into two distinct phases. In the planning phase, the agent formulates a high-level strategy by breaking down the overarching task into smaller, manageable sub-tasks. In the execution phase, the agent performs these sub-tasks sequentially or iteratively, strictly adhering to the initial plan while refining the process based on intermediate results or environmental feedback. Such a design mirrors human problem-solving strategies, offering modularity, scalability, and adaptability for diverse tasks, from software development Qian et al. (2024) to program repair Zhang et al. (2024e) and exception handling Zhang et al. (2024b).

Hierarchy execution is a structured approach where agents decompose complex tasks into smaller, manageable subtasks organized hierarchically. While the core principle of decomposing tasks into subtasks and refining them when needed remains consistent across implementations, structural details and execution strategies vary significantly among frameworks. For example, SELA uses tree-based hierarchies and Monte Carlo Tree Search (MCTS) to optimize AutoML workflows Chi et al. (2024), CodeTree employs explicit task decomposition to identify and evaluate coding strategies with execution feedback for dynamic optimization Li et al. (2024d), and LATS integrates MCTS into a language-agent-driven framework using model-driven value functions

and self-reflection Zhou et al. (2023). Meanwhile, MapCoder and AGILECODER introduce collaborative multi-agent systems for task decomposition Islam et al. (2024a); Nguyen et al. (2024), Data Interpreter uses dynamic graph-based hierarchies for flexible task management Hong et al. (2024), VisionCoder adopts role-based decomposition mirroring traditional software engineering workflows Zhao et al. (2024b), and ScienceAgentBench and Self-Organized Agents incorporate feedback-driven planning to refine subsequent cycles Chen et al. (2024c); Ishibashi & Nishimura (2024).

### 3.2.3 Summary of Execution Structure

| Framework | Tree | Graph | Role | Dynamic Adjust | Feedback | Multi-Agent | MCTS |
|---|---|---|---|---|---|---|---|
| SELA | ✓ | | | | | | ✓ |
| VisionCoder | | | ✓ | | | | |
| AGILECODER | | | ✓ | ✓ | | | |
| MASAI | | | ✓ | ✓ | | ✓ | |
| Data Interpreter | | ✓ | | ✓ | | | |
| MapCoder | ✓ | | | | ✓ | ✓ | |
| CodeTree | ✓ | | | ✓ | | | |
| ScienceAgentBench | | | | | ✓ | | |
| LATS | ✓ | | | | ✓ | | ✓ |

Table 4: Summary of Hierarchy Planning in Different Frameworks. The columns represent the type of planning structure employed, including Tree-Based, Graph-Based, Role-Based, Dynamic Adjustment, Feedback-Driven, Multi-Agent Collaboration, and Monte Carlo Tree Search (MCTS). A checkmark (✓) indicates the framework supports the corresponding structure.

This section provides a summary of the execution structures employed by LLM-based agents, highlighting detailed execution dimensions including task execution, task routing, user interaction, and error handling. These execution dimensions significantly affect how agents adapt to dynamic environments and handle complex tasks. Below are the key execution dimensions with specific examples from §3 of the survey.

**Execution flexibility**. Execution flexibility describes how agents handle task allocation during execution, ranging from static to dynamic allocation:

1. Static Execution: In this case, task allocation is predefined before execution and remains unchanged throughout the process. An example is the Data Director Hong et al. (2024) framework, where agents follow a fixed workflow with predefined tasks and are not dynamically adjusted during the execution.

2. Dynamic Execution: The task allocation changes during execution as the agent receives feedback or as the environment evolves. EvoMAC Hu et al. (2024b) employs a dynamic execution strategy where agents adjust their internal structure, adding or removing agents based on real-time feedback and task complexity.

3. Hybrid Execution: This approach combines both static and dynamic task allocation. For example, in AutoML-GPT Zhang et al. (2023a), task allocation is primarily managed by the central controller but can dynamically adjust based on the results from specialized agents performing tasks like hyperparameter tuning or model design.

**Task routing**. Task routing governs how tasks are passed between agents in a multi-agent system:

1. Rule-based Routing: Tasks follow a specific rule or sequence to move from one agent to another. For instance, in MASAI Arora et al. (2024), tasks follow predefined rules based on the nature of the task, with each agent handling tasks according to a strict order.

2. Agent-based Routing: In this type, one central agent controls the flow of tasks, deciding which agent will handle each task. An example is found in AutoML-GPT Zhang et al. (2023a), where the central LLM oversees task assignment and ensures tasks are routed to the appropriate agents based on the task requirements.

3. Role-based Routing: Agents handle tasks according to their predefined roles. In MAGIS Tao et al. (2024), for instance, different agents like Manager, Developer, and QA Engineer assume roles in the task flow, with each agent taking task when its role is responsible for the incoming part.

**User interaction**. User interaction defines the extent to which users are involved in task execution:

1. Fully-auto: In this case, no user intervention is required. For example, AutoCodeRover Zhang et al. (2024e) operates entirely automatically, with no need for human input during task execution.

2. Human intervene: User interaction is frequent, and users must intervene with the system regularly. MAP-Coder Islam et al. (2024a) requires user feedback to ensure that agents are following the correct task path, especially in error-prone stages of task execution.

3. Hybrid: Some systems balance user input with autonomous execution. In MetaGPT Hong et al. (2023), for instance, the system functions autonomously but allows users to step in for oversight or specific corrections when needed, such as when a complex decision-making process requires a human touch.

**Plan execution**. This dimension describes whether planning and execution are integrated or separated:

1. Plan-Execution in One: One agent handles both planning and execution. In EvoMAC Hu et al. (2024b), for example, the same agent can dynamically plan the next steps and execute them without relying on separate agents for each task.

2. Plan-Execution Separate: The planning and execution roles are handled by separate agents. Parsel Zelikman et al. (2023), for example, separates the planning phase (where tasks are decomposed and strategized) from the execution phase (where the steps are carried out by different agents based on the plan generated).

**Task decomposition**. Task decomposition determines how tasks are broken down and managed:

1. Not Decomposed: In some systems, tasks are handled in their entirety without decomposition. AutoCodeRover Zhang et al. (2024e) may process some high-level tasks as a whole without further division, especially in simple workflows

2. Vertical Decomposition: Tasks are broken down hierarchically, with agents responsible for different levels of the task. This is seen in EvoMAC Hu et al. (2024b), where tasks are decomposed into sub-tasks at various levels, with parent agents overseeing child agents performing specific subtasks

3. Horizontal Decomposition: Tasks are divided into equal parts that are handled concurrently by multiple agents. MASAI Arora et al. (2024) employs horizontal decomposition, dividing a task into multiple smaller subtasks which is handled one following another.

4. Hybrid Decomposition: Some systems use a combination of vertical and horizontal decomposition. AutoML-GPT Trirat et al. (2024), for example, combines both hierarchical task breakdowns (for overseeing large workflows) and parallel execution (for handling repetitive tasks such as data preprocessing)

**Error handling**. Error handling determines how a system deals with problems encountered during execution:

1. Solve in Next Steps: Errors are handled in subsequent steps, either by a different agent or in the following phase of execution. For instance, AutoML-GPT Trirat et al. (2024) might handle errors in model design or tuning by adjusting parameters in later steps of the pipeline.

2. Traceback: The error is traced back to the previous steps to regenerate or correct the output. In MASAI Arora et al. (2024), if an error occurs during task execution, the system might trace back to earlier stages of the task to identify and resolve the root cause before continuing with execution.

### 3.3 External Knowledge

In this section, we summarize current methods for external knowledge acquisition in LLM-based agents. While pretrained LLMs have extensive internal knowledge, external sources are often needed to address out-

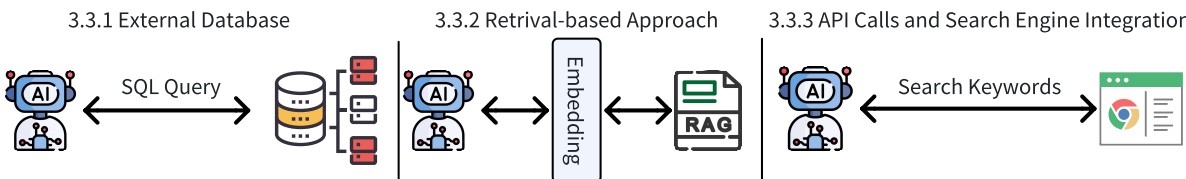

Figure 12: Overview of common external knowledge sources for DS agents.

dated or domain-specific information. We specifically discuss external databases, retrieval-based approaches, API calls, and search engine integration, as well as hybrid methods that combine these techniques.

### 3.3.1 External Knowledge Methods

**External Database** External databases are collections of organized information stored independently of LLMs, designed to serve as reliable sources of well-defined external knowledge for LLM-based agents Gu et al. (2024); Zhang et al. (2023a); Li et al. (2024e;f); Jing et al. (2024); Liu et al. (2024b); Pietruszka et al. (2024); Hassan et al. (2023); Chen et al. (2024a). For example, some utilize historical logs and past experimental results as a structured knowledge base Liu et al. (2024b); Xu et al. (2025), while Hassan et al. (2023) incorporates user-provided datasets with structured databases to enhance contextual understanding. This method provides structured and consistent data, particularly useful for domain-specific tasks that require precision and stability.

**Retrieval-based Approach** Beyond external databases, LLM-based agents employ different retrieval-based approaches to dynamically obtain external knowledge from unstructured sources. For example, Tang et al. (2023) leverages the BM25 retriever, which uses ranking search to identify the most relevant code and documentation based on word frequency and importance, to extract relevant segments based on given instructions. Meanwhile, Li et al. (2024c) uses RAG(Retreival-Augmented Generation) to improve response accuracy and reduce hallucination by retrieving relevant external data and integrating it into the generation process. Furthermore, Guo et al. (2024a) extends Case-Based Reasoning, which retrieves and adapts past cases rather than just ranking or generating text, and Cao et al. (2024) employs LlamaIndex, a data framework for RAG, to efficiently structure, retrieve, and inject knowledge into LLMs. Retrieval-based approaches, especially RAG, are widely used in applications, such as bias detection Li et al. (2025), geospatial analysis Chen et al. (2024b), and financial forecasting Yang et al. (2024a), supporting broader contextual understanding and enhancing adaptability in handling structured and unstructured information.

**API Calls and Search Engine Integration** Another widely adopted approach involves direct interaction with external repositories and search engines, allowing LLM-based agents to directly interact with external repositories or retrieve real-time data from the internet. For instance, Liao et al. (2024) integrates API calls to handle time series analysis by retrieving Prophet models, a statistical forecasting tool that models trends and seasonal variations in time-series data, while Bogin et al. (2024) accesses GitHub repositories and datasets from platforms like Hugging Face. By enabling agents to retrieve external knowledge on demand, API calls and search engine integration provide critical flexibility and responsiveness in dynamic environments.

**Hybrid Approach** To enhance knowledge acquisition, many systems adopt hybrid approaches by combining multiple methods. A common strategy is integrating both API calls and external databases to agents' accessible knowledge sources Merrill et al. (2024); Huang et al. (2024c); Li et al. (2024b); Zhang et al. (2023b); Luo et al. (2024). For example, Merrill et al. (2024) employs Google Search API alongside anonymized wearable health data for retrieving relevant health information. Moreover, some works combine retrieval approaches with an external databases to equip the system with expert-level knowledgeOu et al. (2025). Additionally, several works combine API calls and search engines with retrieval-based approaches for dynamic retrieval Grosnit et al. (2024); Chen et al. (2024b); Tang et al. (2023), while others adopt a fully hybridized approach incorporating all three strategies Guo et al. (2024a); Cao et al. (2024). For instance, Yang et al. (2024a) utilizes third-party APIs for financial data retrieval, applies RAG for financial sentiment analysis, and manages an external database for knowledge storage and retrieval.

### 3.3.2   Summary of External Knowledge

External database, retrieval-based approach, and external API and search engine integration represent three primary methodologies adopted by LLM-based agents for external knowledge acquisition. External databases provide structured and reliable domain-specific information, ensuring precision and stability for targeted tasks. Retrieval-based approaches dynamically extract relevant segments from sources, enhancing contextual comprehension and adaptability. API calls and search engines deliver real-time and frequently updated data, supporting immediate responsiveness in dynamic scenarios. Collectively, these methods enable LLM-based agents to cover a wide range of external knowledge and, moreover, empower LLMs to gauge, verify, enrich, or even refine their internal knowledge, thereby significantly improving their overall accuracy and reliability of their responses.

### 3.4   Reflection

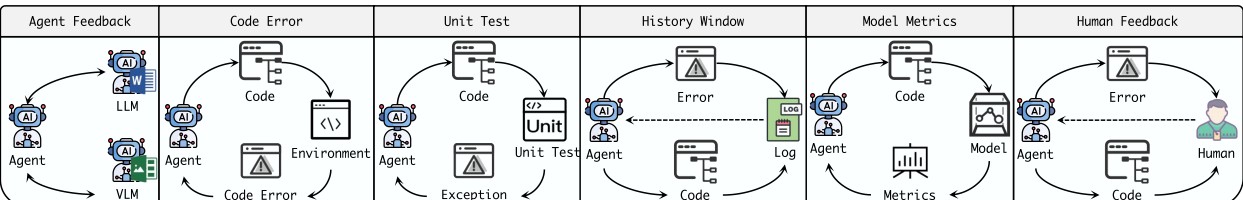

Figure 13: Overview of common reflection techniques in LLM multi-agent systems.

LLM multi-agent systems rely on reflection mechanisms to iteratively refine their performance, enhance robustness, and adapt to complex environments. Reflection, in this context, refers to a system's ability to evaluate its past outputs, identify errors or inefficiencies, and adjust its strategies accordingly—enabling continuous self-improvement.

In this section, we discuss how LLM multi-agent systems employ various reflection mechanisms to improve output quality and system reliability. These mechanisms enable automated agents to iteratively refine their responses based on execution outcomes, predefined evaluation metrics, or external feedback.

### 3.4.1   Reflection Methods

**Agent Feedback:** Many LLM multi-agent systems rely on agent feedback mechanisms, where one or more agents review the outputs of other agents and provide corrective guidance. In code generation tasks, for instance, DA-Code Huang et al. (2024c) , BIASINSPECTOR Li et al. (2025), and AUTOMIND Ou et al. (2025) applies a reviewer agent to evaluate generated scripts, detect syntax or logical errors, and suggest improvements. Similarly, in multimodal tasks, visual language models (VLMs) serve as reviewers to assess image-based outputs for correctness and coherence. MatPlotAgent Yang et al. (2024b) exemplifies this by evaluating visualizations generated from inputs.

**Code Error Handling:** Automated error-handling mechanisms are essential for ensuring reliability of systems that generate and execute code, such as BudgetMLAgent Gandhi et al. (2024), WaitGPT Xie et al. (2024), and DatawiseAgent You et al. (2025). These mechanisms monitor execution failures, capture error messages, and diagnose potential causes. Upon detecting an error, systems analyze faulty outputs and refine code iteratively without external intervention. This allows systems to mimic human programmers—reflecting on errors, identifying root causes, and progressively improving the solution.

**Unit Testing:** Unit testing is a structured validation mechanism where an agent generates test cases and evaluates whether the system's output meets predefined functional requirements. If a test fails, the agent refines the output and reruns the tests iteratively until all test cases pass. This method is widely used in LLM-driven programming tasks, where models generate executable code Guo et al. (2024a). By automatically verifying whether the generated code meets expected functionality, unit testing helps to detect syntax errors, logical flaws, and compatibility issues before execution.

**Model Metrics Feedback:** In tasks with clear quantitative performance indicators, model metrics feedback enables systematic, data-driven optimization. Rather than relying on external evaluations, systems refine their outputs using predefined performance thresholds, such as accuracy, F1 score, or loss reduction. Some implementations use threshold-based optimization, iteratively revising until desired metrics are met. For example, FinRobot Yang et al. (2024a) uses a composite scoring system that integrates normalized performance metrics with weighted evaluation criteria to select or fine-tune models until the target score is achieved. Others adopt exploratory search strategies, such as Monte Carlo Tree Search (MCTS) Chen et al. (2024b), to explore different refinement paths and select the most effective one. This structured approach allows LLM multi-agent systems to self-improve efficiently without requiring human or agent-based feedback at each step.

**History Window:** Unlike mechanisms that focus on immediate corrections, history window mechanisms enable long-term learning by maintaining a log of past outputs and errors Hong et al. (2024)Wang et al. (2025a); Seo et al. (2025); Xu et al. (2025). These logs help systems track recurring mistakes and recognize patterns across multiple iterations. Some implementations enhance this capability with checkpointing, periodically saving stable system states Zhao et al. (2024a). If a later refinement degrades performance, the system can revert to a previous checkpoint. By leveraging historical insights, history window mechanisms prevent repeated failures, allowing systems to refine their decision-making over time and avoid ineffective adjustments.

**Human Feedback:** Despite advancements in automated reflection, human feedback remains indispensable in high-stakes applications. For example, TableAnalyst Freimanis & Andersson Rhodin (2024), a Human-in-the-loop system, allows experts to review and refine outputs, particularly when LLM-generated responses require contextual understanding or ethical considerations. Reinforcement learning from human feedback (RLHF) is a prominent example, where human reviewers assess model-generated responses and provide corrective guidance. Although more interactions required, human feedback mechanisms ensure that outputs align with domain-specific expectations and maintain interpretability in critical applications.

### 3.4.2 Summary of Reflection

Beyond specific implementations, reflection mechanisms can be understood in terms of three fundamental dimensions: the **driver**, the **level**, and the **adaptability** of reflection. These dimensions help contextualize the broader implications of reflection in multi-agent systems, providing insights into how different strategies contribute to long-term performance improvements.

**Drivers of Reflection**. Reflection in LLM multi-agent systems is driven by different mechanisms that shape how the system refines its performance. Some systems improve through internal Hong et al. (2024) or external feedback Merrill et al. (2024), either from agents or human reviewers. Others adopt goal-driven approaches, continuously optimizing their outputs based on predefined performance criteria without relying on explicit external feedback Chen et al. (2024b).

1. Feedback-driven reflection operates based on internal or external evaluation and corrective feedback, where agents or human evaluators assess outputs and provide improving guidance. For example, in agent feedback mechanisms, LLM agents critique one another's outputs, engaging in cycles of feedback and revision Trirat et al. (2024). This process is particularly common in multi-agent collaborations where specialized agents, such as debugging agents in code generation tasks Huang et al. (2024c), verify and refine responses. Human feedback mechanisms introduce expert oversight, ensuring model outputs align with qualitative expectations Luo et al. (2024). While feedback-driven reflection allows flexibility and adaptation to dynamic environments, it also introduces challenges such as response latency and inconsistency in external evaluations.

2. Goal-driven reflection, in contrast, follows a more quantitative optimization approach, where systems refine their outputs based on predefined performance criteria rather than external evaluation. Many multi-agent systems employ metric-based optimization, where refinements are guided by quantitative performance indicators such as BLEU scores in machine translation or loss minimization in model training. Some implementations, such as reinforcement learning and self-play strategies, enable systems to optimize through iterative self-improvement. For instance, AlphaGo Granter et al. (2017) refines its strategies

without relying on external critique by playing against itself. The key advantage of goal-driven reflection lies in its predictability and efficiency, allowing systems to make systematic progress without requiring continuous human or agent-based feedback loops. However, it also risks overfitting to specific metrics, which can reduce generalization and overlook qualitative aspects of task performance.

**Levels of Reflection**. Reflection does not operate uniformly across all components of a system; rather, it varies in scope, with some mechanisms focusing on fine-grained, localized improvements, while others engage in system-wide analysis and optimization.

1. Local reflection focuses on refining individual task iterations, ensuring immediate performance improvements. Common techniques include unit testing Guo et al. (2024a), execution error diagnosis Zhang et al. (2023b), and targeted debugging routines Huang et al. (2024c). The primary advantage of local reflection lies in efficiency, as it enables rapid refinement without requiring the system to analyze historical data or restructure workflows. Additionally, it enhances precision by addressing specific issues within isolated tasks, preventing minor errors from propagating. However, because these mechanisms operate in isolation, they may fail to detect systematic inefficiencies. For example, in code generation Liao et al. (2024), a debugging agent may repeatedly fix syntax errors in isolated function calls without recognizing an underlying flaw in the model's broader logic generation capabilities. As a result, errors may be corrected in the short term but persist across different tasks.

2. Global reflection analyzes patterns across multiple iterations, extracting insights that guide long-term optimization. History window mechanisms exemplify this approach by enabling systems to track recurring mistakes and refine their responses accordingly Bogin et al. (2024). Some implementations further enhance global reflection through checkpointing Hong et al. (2024), where stable system states are periodically saved. If a newer version fails to improve performance, the system can revert to a prior state instead of reinforcing ineffective updates. This method is particularly valuable in preventing cyclical failures, such as a conversational agent repeatedly generating redundant responses due to misaligned reinforcement signals. However, implementing global reflection requires sophisticated memory management and comes with higher computational costs, making it more suitable for large-scale, complex multi-agent systems.

**Adaptability of Reflection**. Another essential consideration is the degree of flexibility and adaptability within the reflection process. While some methods follow fixed, predefined correction strategies, others adjust dynamically based on performance patterns observed during execution.

1. Structured reflection mechanisms operate according to fixed evaluation criteria, ensuring a stable and repeatable refinement process. This category includes unit testing, where outputs must pass predefined test cases Guo et al. (2024a), and threshold-based metric optimization Yang et al. (2024a), where refinements continue until performance metrics such as accuracy or loss reach a set threshold. The primary benefit of structured reflection is its predictability, making it particularly useful in well-defined problem spaces requiring strict correctness—such as automated hyperparameter tuning in machine learning, where models are optimized based on predefined performance metrics. However, its rigidity limits adaptability, making it unsuitable for handling unexpected edge cases or unstructured, open-ended tasks, where predefined evaluation criteria may not capture qualitative aspects of performance.

2. Adaptive reflection mechanisms dynamically adjust their evaluation strategies based on previous attempts or insights gathered from different parts of the system. Systems employing adaptive agent feedback Hong et al. (2024), self-modifying history windows Qi & Wang (2024), or reinforcement learning refine their reflection processes as they accumulate more information. This flexibility allows LLM multi-agent systems to not only correct mistakes but also optimize their refinement strategies over time. For example, in dialogue systems, such as SageCopilot Liao et al. (2024), an adaptive reflection mechanism may modify its response-generation strategy if users repeatedly indicate dissatisfaction, shifting towards more context-aware replies rather than just fixing specific errors. Despite its advantages, adaptive reflection introduces challenges in complexity and computational overhead. Dynamically evolving strategies require greater processing power and careful tuning to prevent unintended biases or instability in the reflection process.

# 4 Analysis from Data Science Perspective

## 4.1 Data Science Tasks

Data science agents are increasingly being integrated into diverse workflows to automate and optimize data-centric tasks. These tasks can be broadly categorized based on their objectives, this section outlines two representative types of tasks where AI agents are commonly deployed: (i) tasks that focus on building and refining machine learning models, and (ii) tasks centered on the generation of insights and outputs from data.

**Building Machine Learning Models** Building machine learning models is a core objective within data science, focusing on creating predictive, explanatory, or generative models to solve domain-specific problems These tasks typically aim to maximize the accuracy, efficiency and generalizability of the model across datasets by automating key processes such as feature engineering, hyperparameter optimization, and model selection Tang et al. (2023). The automation of ML model building uses iterative workflows, where each stage—data preprocessing, model evaluation, and refinement—is informed by prior outputs, often through multi-agent collaborations or tree-based optimization strategies Chi et al. (2024). Additionally, they are characterized by robust integration of tools , alongside domain-specific libraries tailored for computer vision, NLP, and tabular data analysis Grosnit et al. (2024); Xue et al. (2025). These features enable efficient scaling and adaptation to diverse data environments, providing robust solutions to challenges in domains ranging from financial analysis to biomedical research Yang et al. (2024a); Gandhi et al. (2024); Chen et al. (2024a).

**Output Analysis Tasks** Output analysis tasks focus on extracting, interpreting, and communicating insights derived from data. These tasks aim to generate clear and actionable narratives through visualization, summarization, or benchmarking while prioritizing interpretability and relevance Zhao et al. (2024a). LLM-based agents are often used to annotate visualizations and generate and refine textual explanations. For example, output analysis tasks leverage sophisticated tools like Vega-Lite for visualization and natural language models for insight generation, to enhance communication of data-driven findings Xie et al. (2024). In addition, agents are used for data storytelling through automated animated videos that transform raw data into engaging narratives by coordinating visual, textual, and auditory elements Shen et al. (2024).

Analysis tasks such as data cleaning output evaluation focus on quality dimensions like completeness, accuracy, and consistency Li et al. (2024f). Agents typically employ rule-based metrics or statistical heuristics to assess whether cleaned datasets meet project-specific requirements Li et al. (2024f). These evaluations often integrate automated validation pipelines, where agents verify the success of cleaning operations (e.g., deduplication, missing value imputation) and provide reports summarizing detected issues and corrective actions taken Huang et al. (2024c).

## 4.2 Data Science Loop

The data science loop serves as a structured blueprint for how LLM-based agents can systematically enhance and automate key phases of data workflows. Each step from initial data retrieval and cleaning to advanced statistical analysis, model development, and visualization can be augmented by LLM-based agents to improve the efficiency and quality of decision-making. As shown in Figure 14, this loop captures the full lifecycle of a data-driven project and provides a framework for understanding where agent-based systems integrate into and optimize the process.

### 4.2.1 Data Preprocess

The data preprocessing phase includes steps for getting, cleaning, and preparing data. It starts with collecting data from different sources like databases, APIs, web scraping, sensor logs, and research repositories Guo et al. (2024a). Structured data comes from relational databases using SQL, while unstructured text is gathered through APIs or web crawlers Huang et al. (2024c). Some systems combine different types of data, such as tables, images, audio, or text Luo et al. (2024); Cao et al. (2024). Automated pipelines use retrieval-augmented generation (RAG) to find and pull useful data, making sure that it is high quality and relevant for later steps. Li et al. (2024h); Sun et al. (2024a); Rasheed et al. (2024); Yang et al. (2024a).

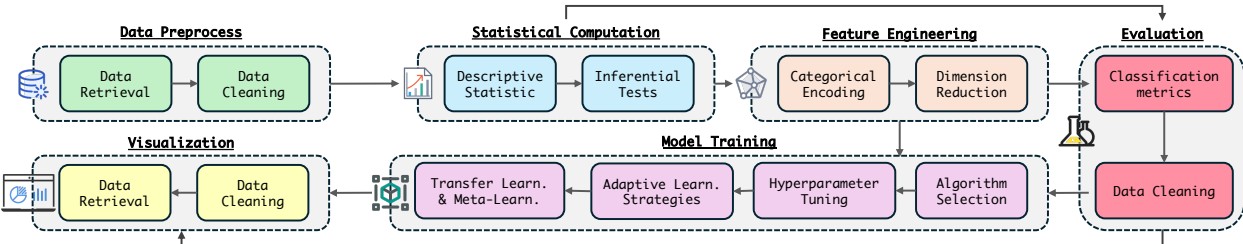

Figure 14: Typical data science loop

After collection, the data is cleaned to improve accuracy and consistency. This step fixes missing values, duplicates, outliers, and inconsistencies that could cause problems in later analysis. If important values are missing, agents will fill gaps or remove incomplete rows based on their domain knowledge and pre-defined rules. Duplicates are found and removed using hashing or fuzzy matching methods like Levenshtein distance for text. Outliers are detected using basic statistical methods like Z-score and IQR filtering, while more advanced techniques like isolation forests and autoencoders handle complex cases. Hong et al. (2024); Li et al. (2024h).

### 4.2.2 Statistical Computation

The statistical computation phase uses statistical methods to analyze data, find patterns, and show relationships. These methods help understand distributions and correlations in the data. Basic techniques include descriptive statistics (mean, median, variance), inferential tests (e.g., t-tests, chi-square tests), and correlation analysis are widely employed to establish baselines and validate data integrity Gu et al. (2024). Methods like hypothesis testing and parametric or non-parametric methods are also utilized to derive insights under uncertainty Zhang et al. (2023b). Parallelized frameworks like Dask or Spark are used to improve computational efficiency for large-scale or distributed datasets Jing et al. (2024).

### 4.2.3 Feature Engineering

In the feature engineering phase, raw data is transformed into meaningful representations that improve model performance. This phase involves creating, selecting, and refining features to ensure they are both relevant and discriminative for the underlying predictive task Pietruszka et al. (2024). A well-designed feature engineering process enables machine learning models to capture patterns effectively while mitigating overfitting and reducing noise. The core techniques in feature engineering include handling missing values, categorical encoding, numerical transformations, and dimensionality reduction. Standard approaches such as one-hot encoding and label encoding allow categorical variables to be numerically represented, while scaling techniques like min-max normalization and standardization ensure consistent feature magnitudes Tang et al. (2023). Advanced feature engineering techniques involve feature construction and selection strategies. Polynomial feature expansion, interaction terms, and domain-specific transformations (e.g., log transformations for skewed distributions) enhance a model's ability to capture complex relationships. Additionally, methods like Principal Component Analysis (PCA) and t-Distributed Stochastic Neighbor Embedding (t-SNE) facilitate dimensionality reduction, improving computational efficiency and reducing redundant information. Feature importance techniques, such as SHAP (SHapley Additive exPlanations) and permutation importance, guide the selection of the most predictive variables Chi et al. (2024).

### 4.2.4 Model Training

In the model training phase, refined data and features are input into machine learning algorithms to create predictive models. This phase encompasses algorithm selection, hyperparameter tuning, and iterative validation to optimize performance. Commonly used algorithms range from traditional methods like linear regression and decision trees to modern neural architectures designed for high-dimensional and multi-modal data Huang et al. (2024a). Agents in training pipelines integrate and use ML libraries and tools to facilitate model creation, while hyperparameter optimization frameworks such as Optuna and Ray Tune enhance the

search for optimal configurations Liu et al. (2024b). Real-time monitoring and adaptive learning strategies are often employed to refine models, particularly in dynamic environments where data evolves over time Chen et al. (2024b). Advanced techniques, such as transfer learning and meta-learning, enable models to leverage knowledge from pre-trained networks, reducing training time and improving performance Luo et al. (2024). The model training phase is iterative by nature, with feedback loops that refine both the model and its underlying assumptions, ensuring robustness and generalizability across unseen data Zhang et al. (2024d).

### 4.2.5 Evaluation

The evaluation phase is to assese the performance and reliability of machine learning models. This step involves the use of metrics tailored to the task at hand, such as accuracy, precision, recall, and F1 score for classification tasks, or RMSE (Root Mean Squared Error) and MAE (Mean Absolute Error) for regression analyses Li et al. (2024e). For unsupervised tasks, metrics like silhouette score and Davies-Bouldin index are employed to evaluate clustering quality Jing et al. (2024). Cross-validation techniques, such as k-fold or leave-one-out, are widely used to estimate model performance on unseen data, ensuring that results generalize beyond the training set Chen et al. (2024c). The integration of automated evaluation frameworks allows for streamlined reporting and comparison, enabling iterative improvement and enhancing the credibility of the final model Xie et al. (2024).

### 4.2.6 Visualization

The visualization phase turns data into easy-to-understand images that help with decision-making. Clear visuals change complex data into simple forms like charts, plots, and dashboards. These help people see patterns, trends, and unusual points. This step uses both fixed and interactive visuals, letting users explore the data in different ways.

Modern data science agents leverage a variety of visualization libraries and tools to create visualizations tailored to different analytical needs Hong et al. (2024); Li et al. (2024h); Liao et al. (2024). In addition to traditional visualizations like line plots and histograms, more advanced visualizations, such as model interpretability plots (e.g., SHAP values), are used to explain model outputs in a transparent manner.

### 4.3 Summary of Data Science Perspective

**Fixed Data Science Pipeline** Data science agents typically follow structured workflows in different stages such as data preprocessing, feature selection, and model trainingYou et al. (2025). LLM-based agent systems like Qi & Wang (2024) and Li et al. (2024f) automate data preprocessing strategies and address data quality issues like missing values, inconsistencies, and duplications. Feature selection mechanismsLi et al. (2024b)further enhance model performance by identifying the most data features, thus optimizing capabilities. Additionally, advanced agent systems, such as Chi et al. (2024); Trirat et al. (2024), dynamically generate optimized data science pipelines.

**Self-planning Training** Data science agents exhibit self-planning training capabilities, autonomously selecting optimal algorithms, configuring hyperparameters, and iteratively validating model performance during the training phaseSeo et al. (2025). This methodology parallels traditional Neural Architecture Search (NAS), an automated process that optimizes neural network designs, but significantly leverages agent-driven decision-making frameworksLiu et al. (2024b). By systematically exploring and refining model configurations based on ongoing feedback, these agents ensure robust and adaptive performance tailored to specific contexts.

**Metrics-based Feedback** In contrast to traditional coding-oriented agents, data science agents utilize rigorous model-based metrics feedback for iterative improvement. FairOPTJung et al. (2025) leverages quantitative metrics such as accuracy, precision, recall, and F1 scores to assess model outputs and systematically enhance their performance through structured refinement cycles. This metrics-driven feedback mechanism

ensures continuous performance enhancement and enables agents to maintain consistent accuracy across diverse analytical scenarios.

**Visualization Analysis** Data science agents distinctly emphasize the visualization of analytical outcomes, recognizing its critical role in data interpretation and communication. Visualization-focused systems Zhao et al. (2024a) generate clear, insightful visual representations that facilitate a comprehensive understanding of complex analytical results. Furthermore, specialized agents like MatplotAgent Yang et al. (2024b) employ visualization explicitly as an informative feedback mechanism for detecting and correcting analytical errors. By integrating visual feedback into their error-handling processes, these agents significantly enhance the interpretability, accuracy, and overall effectiveness of data-driven decision-making.

## 5 Benchmark

A wide range of benchmarks has been developed to evaluate LLM-based data science agents, each reflecting different assumptions about what "data science competence" entails. Table 5 summarizes the major suites. General-purpose benchmarks such as ML-Bench Tang et al. (2023) and DSBench Jing et al. (2024) assess core analytical skills across preprocessing, modeling, feature engineering, and multimodal reasoning. Other benchmarks specialize in particular workflow stages, such as visualization in MatPlotAgent-Bench Yang et al. (2024b) or geospatial analysis in GeoAgent-Bench Liao et al. (2024). Domain-focused benchmarks, such as FoodPuzzle Huang et al. (2024b), GenoTEX Liu & Wang (2024), or AgentClinic Schmidgall et al. (2024), measure scientific and data-driven reasoning under discipline-specific constraints. In addition, TheAgent-Company Xu et al. (2024) provides a practical evaluation framework covering end-to-end automation tasks in corporate environments, including data science, engineering, and business operations.

While these benchmarks provide broad coverage and standardized evaluation settings, they also exhibit structural limitations that may mischaracterize an agent's true capabilities. First, most datasets evaluate short, isolated tasks. Suites such as DSBench or ML-Bench typically test a single operation (e.g., computing a statistic, performing a small transformation, or fitting a lightweight model), thus missing the errors accumulated across multi-stage workflows. Second, rigid task formats allow models to exploit surface regularities rather than reasoning. In MatPlotAgent-Bench, for example, many tasks share highly similar visualization templates (e.g., generating a 2×3 grid of boxplots), so an agent may match the template instead of demonstrating genuine analytical understanding and reasoning. Third, benchmark inputs are often clean, complete, and unambiguous. But LLM agents frequently fail when inputs are messy or underspecified (e.g., irregular schemas or missing values), a scenario that current benchmarks like GeoAgent-Bench still rarely capture.

Each benchmark therefore captures only a particular slice of the capabilities required for real data science practice. General-purpose suites test broad coverage but not long-horizon robustness; workflow-specific and domain-specific benchmarks evaluate deeper skills but in narrower settings. A comprehensive evaluation therefore requires viewing performance across multiple benchmarks, each revealing different aspects of reliability, reasoning, and robustness.

## 6 Future Research Opportunity

Building on the limitations discussed, several research opportunities emerge for improving the reliability of LLM-based data science agents. The core challenges of current systems lie in their vulnerability to hallucination, tendency to generate fragile and non-reproducible code, difficulty maintaining context over long-horizon workflows, and reliance on rigid agent pipelines. The following subsections outline three promising directions, each targeting a different aspect of these limitations and suggesting how next-generation agent systems may overcome them.

### 6.1 Data-Centric Diagnostics

As discussed earlier, many high-impact failures in LLM-based data science agents originate not from the agent's reasoning, but from properties of the data itself. Agents frequently misinterpret irregular schemas, treat categorical columns as numerical, overlook missing-value structures that distort statistical estimates,

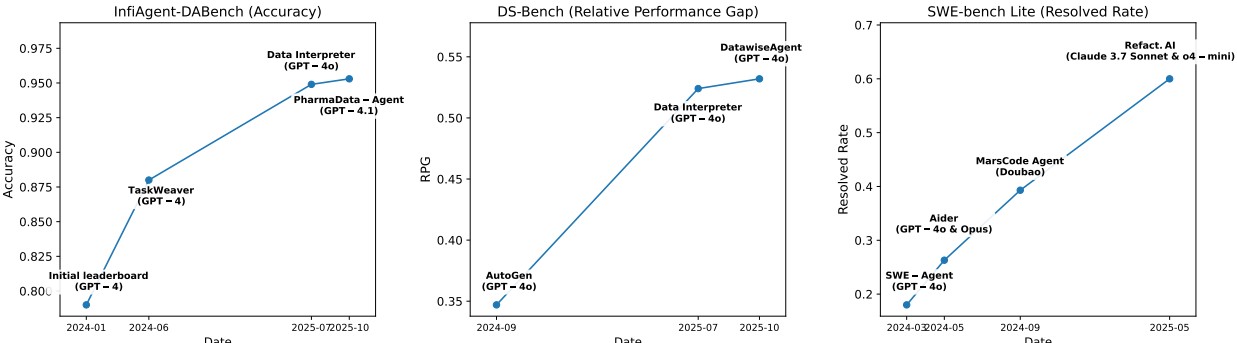

Figure 15: Temporal performance trends across three benchmarks. The figure shows the evolution of state-of-the-art (SOTA) performance on InfiAgent-DABench (accuracy), DS-Bench (relative performance gap, RPG), and SWE-bench Lite (resolved rate). Each data point corresponds to the best-performing system reported at the time of publication and is annotated with the system name and backbone LLM.

or apply methods incompatible with the underlying distribution. Current reflection mechanisms are blind to such data issues: they only inspect the agent's generated text or code, leaving foundational data errors undetected. This limitation explains why agents often produce "correctly reasoned" but fundamentally invalid analytical results. Future research should therefore develop data-centric diagnostic modules that automatically evaluate schema consistency, detect anomalous or distribution-shifted subsets, identify type or semantic mismatches, and trace how specific data defects influence downstream modeling outcomes. By elevating data quality analysis to a first-class reflective capability, agents would be better equipped to prevent systematic failure modes that current systems cannot even observe.

## 6.2 Uncertainty-Aware Workflow Planning

Another highlighted limitation is that current LLM-based agents operate without any awareness of uncertainty. When the model is unsure about a column's meaning, a user's intent, or the correctness of an intermediate step, its downstream behavior does not change and the uncertain output carries the same operational consequences as a confident one. Because agents lack mechanisms to detect low-confidence reasoning, they do not pause execution to request clarification, reconsider assumptions, expand the search space, or branch the workflow. As a result, ambiguous or underspecified steps often trigger confident hallucinations that steer the entire pipeline in an incorrect direction, with errors compounding across subsequent stages. Future research may therefore focus on developing uncertainty-aware workflow planning, where agents explicitly model their confidence and adapt their behavior accordingly. Such systems could flag uncertain decisions, trigger query refinement, invoke stronger verification modes, or explore multiple candidate workflows when ambiguity is high. Introducing uncertainty as a first-class signal would allow agents to dynamically modulate autonomy and mitigate the error-propagation patterns that are pervasive in long-horizon data science tasks.

## 6.3 Pipeline-Level Reflection

A further limitation is that existing reflection mechanisms operate only at the step level. Reviewers check the latest code snippet, self-reflection revises the previous response, and unit tests validate single functions. Yet data science workflows are inherently multi-stage, and small upstream issues, such as incorrect filtering, misaligned merges, omitted preprocessing, often propagate silently into feature engineering, model training, and evaluation. Current agents lack the ability to revisit earlier stages, attribute downstream errors to upstream decisions, or assess the global coherence of the entire analytical pipeline. Future research may address this gap by developing pipeline-level reflection: constructing explicit workflow graphs, tracing how information and errors flow across steps, identifying high-risk nodes, performing stage-level sanity checks, and enabling global revisions that correct the pipeline holistically rather than locally. Such mechanisms would transform reflection from shallow post-hoc editing into a principled, system-level reliability process.

| Benchmark | Source | # of Tasks | Task Types |
|---|---|---|---|
| BLADE Gu et al. (2024) | Literature | 714 | multiple-choice and ground truth-analysis studies |
| InfiAgent-DABench Hu et al. (2024a) | GitHub | 257 | Data Analysis and ML |
| ML-Bench Tang et al. (2023) | GitHub | 9641(168) | Multimodal, time series, audio, LLM, vision, biomedical, graphs |
| DevBench Li et al. (2024a) | Github | 22 | DL, CV, and NLP |
| SUPER Bogin et al. (2024) | GitHub | 799 | Research data science challenges |
| DA-Code Huang et al. (2024c) | Kaggle, GitHub | 500 | Data wrangling, EDA, ML |
| FeatEng Pietruszka et al. (2024) | Kaggle | 103 | Classification, regression, feature engineering |
| Tapilot Li et al. (2024e) | Kaggle | 1024 | Data analysis, request clarification |
| MLE-Bench Chan et al. (2024) | Kaggle | 75 | Modelling Tasks(image, video, LLMs, tabular) |
| DSBench Jing et al. (2024) | ModelOff, Kaggle | 540 | Data analysis, modeling |
| PyBench Zhang et al. (2024c) | Kaggle, Arxiv, multimedia files | 143 | Data Analysis, ML, image, text, and audio analysis |
| DSEval Zhang et al. (2024d) | Tutorials, StackOverflow, Kaggle | 825 | Data analysis |
| Spider2-V Cao et al. (2024) | Tutorials, enterprise applications | 494 | Warehousing, transformation, visualization |
| MatPlotAgent-Bench Yang et al. (2024b) | Matplotlib, OriginLab | 100 | Standard and advanced visualization |
| ScienceAgentBench Chen et al. (2024c) | Peer-reviewed publications | 102 | Data processing, modeling, visualization |
| SagePilot Liao et al. (2024) | External datasets | 276 | SQL-related tasks |
| Text2Analysis He et al. (2024) | LLM and human-generated | 2249 | Data analysis, modeling |
| DataNarrative Islam et al. (2024b) | Pew Research, Tableau Public, GapMinder | 1449 | Data-driven visualization, storytelling |
| Insigt-Bench Sahu et al. (2024) | ServiceNow platform | 100 | Data Analysis tasks |
| MMAU Yin et al. (2024) | In-house, Kaggle, DeepMind-Math | 20 | DS/ML, contest-level coding, math |
| GeoAgent-Bench Liao et al. (2024) | GitHub, tutorials, LLM generation | 19,504 | Single-turn, multi-turn in Geo-spatial analysis |
| MLAgentBench Huang et al. (2023) | Kaggle, canonical datasets | 13 | Research Image, text, graph, tabular, time series ML tasks |
| Merrill et al. (2024) | Human experts, wearable data | 4172 | Numerical, open-ended reasoning in personal health |
| AgentClinic Schmidgall et al. (2024) | USMLE, MIMICIV, NEJM, MedAQ | 535 | Patient interaction, multimodal data collection in clinics |
| GenoTEX Liu & Wang (2024) | GEO, TCGA, NCBI Gene Database | 1146 | Dataset selection, processcioning, statistical analysis in Genomics |
| TheAgentCompany Xu et al. (2024) | Company websites, human examples | 175 | DS tasks in company settings |
| FoodPuzzle Huang et al. (2024b) | FlavorDB | 2744 | Molecular prediction and profile completion in flavor science |
| MLGym Nathani et al. (2025) | Publications and datasets | 13 | DS, CV, NLP, RL, and game theory |
| DataSciBench Zhang et al. (2025a) | CodeGeeX, BCB, and human | 519 | Data Analysis, modeling, data Visualization |
| Chen et al. (2024a) | CREEDS and GEO | | GEO Database and Drug Repurposing Database in medical fields |
| BIODSA-1K Wang et al. (2025b) | Publications | 1029 | Biomedical hypothesis and analysis |
| DS-1000 Lai et al. (2023) | Stackoverflow | 1000 | Code generation in DS |
| BioDSBench Wang et al. (2024b) | Published studies, TCGA-type genomics, and clinical data | 293 | Biomedical coding tasks |
| ML-Dev-Bench Padigela et al. (2025) | Unknown | 30 | Data processing, modeling, API integration |
| TimeSeriesGym Cai et al. (2025) | Kaggle, Github, publications, hand-crafted | 34 | Time-series problem |

Table 5: Summary of Benchmarks in Data Science Perspective

# 7 Conclusion

This survey provides a comprehensive analysis of Large Language Model (LLM)-based agents in data science, addressing key aspects from both agent design and data science perspectives. It systematically explores

different agent roles, execution structures, external knowledge acquisition methods, and reflection techniques. The study introduces a dual-perspective framework bridging general agent design principles with specific data science requirements, covering structures from single-agent to dynamic multi-agent systems and execution methods from dynamic planning to static workflows. Overall, this paper offers a comprehensive summary of existing work on LLM-based data science agents.

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
