# OpenReview forum: "Large Language Model-based Data Science Agent: A Survey"
_TMLR — Accepted by TMLR_

### Review · Reviewer_cYJu · 2025-10-18

**Summary Of Contributions:**

This paper presents a comprehensive and well-structured survey of Large Language Model (LLM)-based agents for data science. It does a good job of linking together two major views: how agents are designed (roles, execution, knowledge, reflection) and how they operate in data science workflows (data preprocessing, modeling, evaluation, and visualization). The structure is clear, the coverage of related work is strong, and the dual-perspective framework effectively captures one of the key trends in this area.

**Additional Comments:**

NA

**Audience:**

Yes

**Audience Explanation:**

Yes. The survey targets a timely and rapidly evolving area that aligns closely with TMLR’s audience in machine learning, AI systems, and applied research.

**Claims And Evidence:**

Yes

**Claims Explanation:**

Yes. Although the paper’s claims are solidly supported by abundant references to recent studies, some sections would benefit from stronger analytical framing.

**Requested Changes:**

The paper could be improved in several aspects. First, several subsections—particularly Agent Role Design (§3.1), Execution Structure (§3.2), External Knowledge (§3.3), Reflection (§3.4), and Data Science Loop (§4.2)—should explicitly state the research questions that the surveyed studies aim to address. At the moment, the summaries describe what each work does, rather than explaining why the research was conducted or what gap it was intended to fill. Including such guiding questions would make the discussion more analytical and less descriptive.

Second, there is a noticeable lack of discussion regarding real-world or industry applications. Although the paper summarizes academic advances very well, it does not yet address how these agent systems are being—or could be—deployed in industrial settings.

Lastly, some sections would benefit from mathematical formalization to add precision and scientific rigor. Specifically:
Execution Structure (§3.2) could formalize task planning and coordination (e.g., using MDP or DAG formulations).
Reflection (§3.4) could mathematically articulate iterative feedback and performance optimization.
External Knowledge (§3.3) could include retrieval scoring or hybrid knowledge fusion equations.
Data Science Loop (§4.2) could describe workflow stages using composable operators under a unified optimization objective.

In all, this is a strong and timely survey. With clearer articulation of research questions, expanded discussion of industry relevance, and a math-backed analytical approach, it could become one of the most authoritative analyses of LLM-based data science agents to date.

---

> ### Author Response · Authors · 2025-11-23
> **Response to Requested Change 1 (Lack of research questions)**
>
> We agree that the earlier version of the survey did not clearly articulate the research questions underlying each major component of our framework. To address this, we revised the introduction and added a new paragraph that introduces five interrelated research questions, each corresponding to one of the primary sections of the survey. These research questions serve as the conceptual backbone of the paper, clarifying why we organize the survey in this way and helping readers quickly locate the sections most relevant to their interests. The revised text is as follows:
>
> > A central goal of this survey is to clarify how LLM agents can be organized to perform the functions of a data scientist. This leads to a sequence of increasingly detailed research questions. (RQ1) At the organizational level, how should agents’ roles and responsibilities be defined so that complex analytical tasks can be effectively decomposed? (RQ2) Given these roles, how should each agent carry out its reasoning and tool use to ensure reliable execution? (RQ3) Execution usually depends on information beyond the agent’s internal context. What external knowledge should agents access, and how should such knowledge be integrated into their decisions? (RQ4) Even with access to the necessary knowledge, agents still make mistakes, raising the question of what reflection and self-correction mechanisms are required to detect and repair errors during operation. (RQ5) Finally, data science workflows are inherently iterative, prompting the question of how these agent-level capabilities can support end-to-end, loop-based processes rather than isolated single-step tasks.

---

> ### Author Response · Authors · 2025-11-23
> **Response to Requested Change 2 (Need industrial applications)**
>
> Our earlier version did not sufficiently discuss how the methods relate to real-world or industry applications. We revised Section 3 to clarify this. In every subsection, each technique now includes an explicit description of its advantages, its limitations, and the types of real-world scenarios where it is most appropriate or inappropriate.
>
> In addition, we added a summary table at the end of the section that uses a one-hot style comparison to present the trade-offs across methods. These tables highlight not only their relative capabilities but also their practical applicability in industrial settings. This revision allows readers to make clearer cross-method comparisons and to understand how different agent designs may perform under the constraints of real-world data science workflows.
>
> Take **Section 3.1 Agent Role Design** as an illustrative example:
>
> - In *3.1.1 Single Agent*, we added an ending paragraph:
> > Single-agent designs offer minimal overhead and are easy to deploy, making them attractive for low-latency or low-cost settings. However, since all reasoning, memory, and execution are concentrated in a single model, they inherit most of the LLM limitations: hallucinations are unmitigated, generated code is fragile, and long-horizon tasks often suffer from goal drift and lost context. As a result, single-agent systems are often insufficient for complex or multi-stage workflows that require long-horizon reasoning, consistent state tracking, or robust code execution.
> - In *3.1.2 Two Agents*, the ending paragraph:
> > Two-agent designs improve over single-agent systems by splitting responsibilities into complementary roles. Planner–executor structures provide more structured task decomposition, and coder–reviewer structures introduce lightweight verification, catching some errors or hallucinations that a single agent would miss. However, the approach remains sensitive to failures such as a planner hallucinating an incorrect plan or a reviewer failing to identify subtle logical errors. Communication cost also increases, and the reliability gains depend heavily on how well the two roles align. As a result, two-agent systems offer a moderate balance between simplicity and robustness, but their effectiveness is bounded by the underlying LLM's consistency and self-correction ability.
> - For *3.1.3 Multiple agents* we added:
> > Multi-agent architectures offer the highest scalability and strongest resistance to long-context forgetting, as each agent operates within a narrowly scoped role.
> They can achieve high reliability through division of labor, modular workflows, and cross-agent verification.
> Yet the increased number of agents introduces coordination overhead, susceptibility to miscommunication, and longer execution latency.
> These systems are powerful but harder to control, and their effectiveness depends on careful manual workflow design.
> - For *3.1.4 Dynamic Agents* we added the ending paragraph:
> > Dynamic agent generation offers high flexibility by creating agents on demand rather than relying on manually designed roles, prompts, and pipeline. This allows the system to adjust its structure at runtime and better accommodate diverse or evolving data science tasks. However, dynamically spawning agents also amplifies LLM uncertainty, reduces reproducibility, and increases computational and coordination overhead. As a result, these systems provide strong adaptability but exhibit lower predictability and stability, making them challenging to deploy in stable production settings.
> - In *3.1.5 Summary of Agent Role*, we included a cross-method comparison table:
>
>
> ### Trade-off summary across agent role designs
> **Legend:** ⬤ = strong, ◑ = moderate, ◯ = weak
>
> | **Dimension** | **Single Agent** | **Two-Agent** | **Multi-Agent** | **Dynamic Agents** |
> |---------------|------------------|----------------|------------------|----------------------|
> | **Reliability**          | ◯ | ◑ | ⬤ | ◑ |
> | **Scalability**          | ◯ | ◑ | ⬤ | ⬤ |
> | **Coordination Cost**    | ⬤ | ◑ | ◯ | ◯ |
> | **Predictability/Stabillity**       | ◑ | ◑ | ⬤ | ◯ |
> | **Industrial Applicability** | Quick analyses, ad-hoc queries, and low-risk automation | Medium-scale workflows requiring basic verification and predictable execution | Production pipelines with modular stages, quality checks, and stable data dependencies | Exploratory analytics, quick pipeline prototyping, or changeable environment |

---

> ### Author Response · Authors · 2025-11-23
> **Response to Requested Change 3 (Needs stronger mathematical formalization)**
>
> We agree that mathematical formalization is essential for precisely defining and distinguishing the mechanisms discussed. In the revised version, we strengthened the mathematical rigor in §3.2, §3.3, and §3.4. As an illustrative example, we show a formal comparison between two core feedback mechanisms: model metrics feedback and agent feedback:
>
> **Model metrics feedback**. A single agent adjusts its internal scoring or decision rule to improve a task-specific metric. The objective function can be written abstractly as:
>
> $$
> \min \ \mathbb{E}\left[\mathrm{M}\big(f(x),\, y\big)\right].
> $$
>
> Here, $f(x)$ denotes the agent’s output for input $x$, $y$ is the target answer, $\mathrm{M}(\cdot)$ is any task-level performance metric (e.g., an accuracy gap or evaluation rubric score).
>
> **Agent feedback**. In multi-agent settings, each agent refines its behavior by integrating peer feedback. This process can be formalized as:
>
> $$
> \min \ \mathbb{E}\left[\sum_{j\neq i} w_{ij}\, d\left(f_i(x),\, r_{ij}\right)\right].
> $$
>
> Here, $f_i(x)$ is the output of agent $A_i$, $r_{ij}$ is the feedback or assessment provided by agent $A_j$, $d(\cdot,\cdot)$ is a discrepancy measure, $w_{ij}$ is the confidence weight that agent $A_i$ to the reviewer agent $A_j$.

---

### Review · Reviewer_EwxZ · 2025-10-24

**Summary Of Contributions:**

This survey provides a comprehensive analysis of LLM-based agents specifically designed for data science tasks. The authors propose a dual-perspective framework to structure their review. From the "agent perspective," they examine the fundamental design principles of these agents, including their roles, execution structures, knowledge integration, and reflection mechanisms. From the "data science perspective," they map the application of these agents across the typical data science workflow, such as data preprocessing, model development, evaluation, and visualization. The paper's main contributions are a thorough review of recent advancements in this area and the introduction of a framework that connects general agent design with the practicalities of data science.

**Audience:**

No

**Audience Explanation:**

While this paper is a well-structured and comprehensive survey, it is unlikely to be of significant interest to a substantial portion of the TMLR audience due to a fundamental mismatch in its contribution style and the journal's focus. TMLR's readership primarily seeks original, cutting-edge research that presents novel algorithms, theoretical insights, or rigorous empirical results that advance the field. This paper's central contributions--a review of existing literature and an organizational framework--are inherently descriptive rather than generative. Consequently, it lacks the deep, critical analysis and presentation of new, validated techniques that a research-focused audience expects. A researcher reading TMLR is typically looking for the next breakthrough to build upon, not a retrospective summary of what has already been established. Therefore, despite its quality as a survey, its findings are organizational rather than empirical or theoretical, making it better suited for a dedicated review journal rather than a top-tier venue for original machine learning research.

**Claims And Evidence:**

No

**Claims Explanation:**

Strengths :-

- The survey provides a systematic breakdown of agent design principles. The "Analysis from Agent Perspective" section methodically addresses key components, including agent roles, execution structures, external knowledge, and reflection. For instance, in discussing "Agent Role Design," the paper outlines a clear progression from "single-agent designs, which manage all tasks independently," to more intricate "two-agent systems" and "multi-agent systems". This demonstrates a comprehensive approach to cataloging the current state of agent architecture.
- The document is organized in a way that enhances readability and comprehension. A key feature is "Figure 1: Structure of This Survey," which functions as a visual table of contents, clearly outlining the paper's logical flow for the reader. The authors employ direct and functional prose, making the dense technical subject matter relatively straightforward to follow.


Weaknesses :-
- The survey presents a very optimistic view of LLM-based agents, focusing almost exclusively on their capabilities and potential. There is no substantive discussion of the inherent limitations of current LLMs, such as their propensity for hallucination, their difficulties with complex mathematical reasoning, or the brittleness of their code generation. A more balanced review would have dedicated a section to these challenges and how current research is attempting to mitigate them.

- While the survey is broad, it sometimes lacks technical depth. For instance, in the discussion of "Reflection," the paper lists several techniques like "agent feedback for self-correction, model metrics feedback for optimization, code error handling for reliability, and history window mechanisms for long-term learning". However, it does not delve into the specific algorithms or implementation details of these mechanisms, which would be highly beneficial for a technical audience.

- The paper includes a section on benchmarks and provides an extensive table of existing ones. However, it does not critically analyze the limitations of these benchmarks. For example, it doesn't discuss the potential for "Goodhart's Law" (where a metric ceases to be a good measure once it becomes a target) in the context of LLM agent evaluation or the challenges of creating benchmarks that truly reflect real-world data science complexity.

-  The "Future Research Opportunity" section identifies important areas like "Trainable architecture," "Advanced Reflection Mechanisms," and "Multimodal Processing". However, these points could have been more deeply integrated with the weaknesses and gaps identified throughout the survey. A more impactful conclusion would have directly linked the identified limitations in the current state-of-the-art to these future research directions.

**Requested Changes:**

- Add a dedicated analysis of the inherent limitations of LLM agents (e.g., hallucination, brittleness, computational cost) and the shortcomings of current evaluation benchmarks to provide a more balanced and scientifically grounded perspective.
- Move beyond cataloging different agent designs and reflection mechanisms by providing a comparative analysis of their trade-offs, helping readers understand the relative strengths, weaknesses, and ideal use cases for each approach.
- Revise the conclusion to explicitly link the limitations and gaps identified throughout the survey to the proposed "Future Research Opportunities," framing the future work as a direct and well-motivated agenda to address the field's key challenges.

---

> ### Author Response · Authors · 2025-11-23
> **Response to Requested Changes 1 and 2 (Need deep discussions about limitations and comparisons)**
>
> We agree that our earlier version presented an overly optimistic view of LLM-based agents. We now explicitly acknowledge that relying on a single LLM introduces fundamental limitations, which is why multi-agent systems have become necessary and why we organize around system-design strategies that address these shortcomings. In addition, we add more horizontal comparisons to show the pros, cons, and tradeoffs among different methods.
>
> We conducted the following 4 revisions:

---

> > ### Author Response · Authors · 2025-11-23
> > **Revision 1: Rewriting the introduction of Section 3.**
> >
> > We fully rewrote the opening paragraph of Section 3. Instead of using a generic description, we now motivate the entire section by discussing core limitations of current LLMs (e.g., hallucination, fragile code generation, and context forgetting), and explains why the design dimensions covered in Sections 3.1–3.4 are effective ways to mitigate these issues. This provides clearer motivation for our organization and creates a more natural lead-in to the methods.
> >
> > Revised text:
> >
> > > LLMs provide strong linguistic and generative capabilities, but they exhibit structural limitations that become particularly evident in data science workflows. First, LLMs are prone to hallucination, e.g., fabricating column names, statistical results, or entire processing steps. Second, LLM-generated code is brittle. Even when logically correct in natural language, produced scripts frequently break due to issues such as type inconsistencies, missing dependencies, unstable pipelines, or non-reproducible execution paths. Third, LLMs struggle with long-horizon tasks. Their limited ability to retain and manipulate extended context leads to goal drift, forgotten assumptions, and inconsistencies across multi-step analytical processes. These limitations show that raw LLMs are not reliable as end-to-end data science systems and need structured agent architectures to provide the capabilities they lack and to address common failure modes.
> > Motivated by these limitations, recent work has increasingly turned to agent architectures that introduce structure, modularity, and external support, enabling LLMs to compensate for their inherent limitations. For example, grounding and verification mechanisms help reduce hallucination, controlled execution improves reliability, and explicit role decomposition mitigates long-context instability. Agent systems thus do not merely wrap LLMs—they provide the organizational scaffolding needed to transform a general-purpose model into a dependable data science practitioner.
> > Seen from this perspective, our analysis focuses on four design dimensions: agent role design (3.1, execution structure (3.2), external knowledge integration (3.3), and reflection mechanisms (3.4). They form a structured solution to the inherent limitations of LLMs in data science settings. The remainder of this section examines each dimension in turn, showing how existing systems operationalize these design principles in practice.

---

> > ### Author Response · Authors · 2025-11-23
> > **Revision 2: Adding advantages, limitations, and trade-offs for each method**
> >
> > For every method discussed in Section 3, we added an ending paragraph describing its strengths, weaknesses, and trade-offs. In addition, each subsection now concludes with a one-hot comparison table summarizing the method’s capabilities, limitations, and suitable application scenarios.
> >
> > Take **Section 3.1 Agent Role Design** as an illustrative example:
> >
> > - In *3.1.1 Single Agent*, we added an ending paragraph:
> > > Single-agent designs offer minimal overhead and are easy to deploy, making them attractive for low-latency or low-cost settings. However, since all reasoning, memory, and execution are concentrated in a single model, they inherit most of the LLM limitations: hallucinations are unmitigated, generated code is fragile, and long-horizon tasks often suffer from goal drift and lost context. As a result, single-agent systems are often insufficient for complex or multi-stage workflows that require long-horizon reasoning, consistent state tracking, or robust code execution.
> > - In *3.1.2 Two Agents*, the ending paragraph is:
> > > Two-agent designs improve over single-agent systems by splitting responsibilities into complementary roles. Planner–executor structures provide more structured task decomposition, and coder–reviewer structures introduce lightweight verification, catching some errors or hallucinations that a single agent would miss. However, the approach remains sensitive to failures such as a planner hallucinating an incorrect plan or a reviewer failing to identify subtle logical errors. Communication cost also increases, and the reliability gains depend heavily on how well the two roles align. As a result, two-agent systems offer a moderate balance between simplicity and robustness, but their effectiveness is bounded by the underlying LLM's consistency and self-correction ability.
> > - For *3.1.3 Multiple agents* we added:
> > > Multi-agent architectures offer the highest scalability and strongest resistance to long-context forgetting, as each agent operates within a narrowly scoped role.
> > They can achieve high reliability through division of labor, modular workflows, and cross-agent verification.
> > Yet the increased number of agents introduces coordination overhead, susceptibility to miscommunication, and longer execution latency.
> > These systems are powerful but harder to control, and their effectiveness depends on careful manual workflow design.
> > - For *3.1.4 Dynamic Agents* we added the ending paragraph:
> > > Dynamic agent generation offers high flexibility by creating agents on demand rather than relying on manually designed roles, prompts, and pipeline. This allows the system to adjust its structure at runtime and better accommodate diverse or evolving data science tasks. However, dynamically spawning agents also amplifies LLM uncertainty, reduces reproducibility, and increases computational and coordination overhead. As a result, these systems provide strong adaptability but exhibit lower predictability and stability, making them challenging to deploy in stable production settings.
> > - In *3.1.5 Summary of Agent Role*, we included a cross-method comparison table, including the industrial applicability (suggested by reviewer cYJu):
> >
> >
> > ### Trade-off summary across agent role designs
> > **Legend:** ⬤ = strong, ◑ = moderate, ◯ = weak
> >
> > | **Dimension** | **Single Agent** | **Two-Agent** | **Multi-Agent** | **Dynamic Agents** |
> > |---------------|------------------|----------------|------------------|----------------------|
> > | **Reliability**          | ◯ | ◑ | ⬤ | ◑ |
> > | **Scalability**          | ◯ | ◑ | ⬤ | ⬤ |
> > | **Coordination Cost**    | ⬤ | ◑ | ◯ | ◯ |
> > | **Predictability/Stabillity**       | ◑ | ◑ | ⬤ | ◯ |
> > | **Industrial Applicability** | Quick analyses, ad-hoc queries, and low-risk automation | Medium-scale workflows requiring basic verification and predictable execution | Production pipelines with modular stages, quality checks, and stable data dependencies | Exploratory analytics, quick pipeline prototyping, or changeable environment |

---

> > ### Author Response · Authors · 2025-11-23
> > **Revision 3: Expanding Section 5 on benchmark limitations.**
> >
> > We also rewrote the whole Section 5 to provide a more analytical discussion of the limitations in current benchmark evaluations. The updated text identifies three common issues that may cause benchmark results to mischaracterize an agent’s true capabilities:
> >
> > 1. evaluating only final outputs without accounting for error accumulation across intermediate steps,
> > 2. rigid task formats that encourage pattern learning rather than genuine data understanding and reasoning,
> > 3. problem settings that are overly clean and do not reflect realistic complexity.
> >
> > Revised text:
> > > A wide range of benchmarks has been developed to evaluate LLM-based data science agents, each reflecting different assumptions about what “data science competence” entails. Table 5 summarizes the major suites. General-purpose benchmarks such as ML-Bench and DSBench assess core analytical skills across preprocessing, modeling, feature engineering, and multimodal reasoning. Other benchmarks specialize in particular workflow stages, such as visualization in MatPlotAgent-Bench or geospatial analysis in GeoAgent-Bench. Domain-focused benchmarks, such as FoodPuzzle, GenoTEX, or AgentClinic, measure scientific and data-driven reasoning under discipline-specific constraints. In addition, TheAgentCompany provides a practical evaluation framework covering end-to-end automation tasks in corporate environments, including data science, engineering, and business operations.
> >
> > > While these benchmarks provide broad coverage and standardized evaluation settings, they also exhibit structural limitations that may mischaracterize an agent’s true capabilities. First, most datasets evaluate short, isolated tasks. Suites such as DSBench or ML-Bench typically test a single operation (e.g., computing a statistic, performing a small transformation, or fitting a lightweight model), thus missing the errors accumulated across multi-stage workflows. Second, rigid task formats allow models to exploit surface regularities rather than reasoning. In MatPlotAgent-Bench, for example, many tasks share highly similar visualization templates (e.g., generating a 2×3 grid of boxplots), so an agent may match the template instead of demonstrating genuine analytical understanding and reasoning. Third, benchmark inputs are often clean, complete, and unambiguous. But LLM agents frequently fail when inputs are messy or underspecified (e.g., irregular schemas or missing values), a scenario that current benchmarks like GeoAgent-Bench still rarely capture.
> >
> > > Each benchmark therefore captures only a particular slice of the capabilities required for real data science practice. General-purpose suites test broad coverage but not long-horizon robustness; workflow-specific and domain-specific benchmarks evaluate deeper skills but in narrower settings. A comprehensive evaluation therefore requires viewing performance across multiple benchmarks, each revealing different aspects of reliability, reasoning, and robustness.

---

> > ### Author Response · Authors · 2025-11-23
> > **Revision 4: (Indirect but related) Research Question Proposal**
> >
> > In addition, we revised the Intro section based on Reviewer cYJu’s feedback by adding five research questions that frame the overall structure of the survey. Although this section does not enumerate LLM limitations directly, the questions are tightly connected to the five major sections of the paper and reflect our motivation for organizing system-design methods around the challenges introduced by these limitations.
> >
> > The revised text is as follows:
> >
> > > A central goal of this survey is to clarify how LLM agents can be organized to perform the functions of a data scientist. This leads to a sequence of increasingly detailed research questions. (RQ1) At the organizational level, how should agents’ roles and responsibilities be defined so that complex analytical tasks can be effectively decomposed? (RQ2) Given these roles, how should each agent carry out its reasoning and tool use to ensure reliable execution? (RQ3) Execution usually depends on information beyond the agent’s internal context. What external knowledge should agents access, and how should such knowledge be integrated into their decisions? (RQ4) Even with access to the necessary knowledge, agents still make mistakes, raising the question of what reflection and self-correction mechanisms are required to detect and repair errors during operation. (RQ5) Finally, data science workflows are inherently iterative, prompting the question of how these agent-level capabilities can support end-to-end, loop-based processes rather than isolated single-step tasks.

---

> ### Author Response · Authors · 2025-11-23
> **Response to requested change 3 (Limitation motivated future works)**
>
> We agree that the previous **Future Research Opportunity** section was not sufficiently linked to the limitations identified earlier in the survey. The earlier text did not clearly articulate how the proposed directions were motivated by the challenges discussed throughout the paper.
>
> In the revision, we removed two subsections. The earlier subsection on **Multimodal Processing** did not correspond to any major limitation discussed in the survey, and the idea of **Trainable Architecture** has since been explored by recent work published after we completed the initial drafting of this survey. We therefore eliminated both topics.
>
> The previous subsection on **Advanced Reflection Mechanisms** was also overly broad. Our original intention was to address failures in long-horizon reasoning and error propagation throughout multi-step workflows. To make this focus more precise, we reframed this direction as **Pipeline-Level Reflection**, which more accurately describes the type of long-horizon error localization and correction that current systems lack.
> In addition to refining and narrowing our previous content, we also introduced two research directions that directly reflect the limitations identified throughout the survey and are not addressed by existing work. The first is **Data-Centric Diagnostics**, which targets the data-side failure modes highlighted in Sections 3 and 5, such as schema irregularities, type mismatches, and distributional anomalies that current reflection mechanisms cannot detect. The second is **Uncertainty-Aware Workflow Planning**, motivated by our observation that present-day agents execute entire pipelines without any awareness of confidence or ambiguity, resulting in error propagation when intermediate reasoning steps are unreliable. These two directions respond directly to gaps we identified in current systems and offer practical, unexplored avenues for improving the robustness of LLM-based data science agents.
>
> The revised Section 6 is as follows:

---

> > ### Author Response · Authors · 2025-11-23
> > **Revised text for future works**
> >
> > **6 Future Research Opportunity**
> >
> > Building on the limitations discussed, several research opportunities emerge for improving the reliability of LLM-based data science agents. The core challenges of current systems lie in their vulnerability to hallucination, tendency to generate fragile and non-reproducible code, difficulty maintaining context over long-horizon workflows, and reliance on rigid agent pipelines.
> > The following subsections outline three promising directions, each targeting a different aspect of these limitations and suggesting how next-generation agent systems may overcome them.
> >
> > **6.1 Data-Centric Diagnostics**
> >
> > As discussed earlier, many high-impact failures in LLM-based data science agents originate not from the agent's reasoning, but from properties of the data itself. Agents frequently misinterpret irregular schemas, treat categorical columns as numerical, overlook missing-value structures that distort statistical estimates, or apply methods incompatible with the underlying distribution. Current reflection mechanisms are blind to such data issues: they only inspect the agent's generated text or code, leaving foundational data errors undetected. This limitation explains why agents often produce ``correctly reasoned'' but fundamentally invalid analytical results. Future research should therefore develop data-centric diagnostic modules that automatically evaluate schema consistency, detect anomalous or distribution-shifted subsets, identify type or semantic mismatches, and trace how specific data defects influence downstream modeling outcomes. By elevating data quality analysis to a first-class reflective capability, agents would be better equipped to prevent systematic failure modes that current systems cannot even observe.
> >
> > **6.2 Uncertainty-Aware Workflow Planning**
> >
> > Another highlighted limitation is that current LLM-based agents operate without any awareness of uncertainty. When the model is unsure about a column’s meaning, a user’s intent, or the correctness of an intermediate step, its downstream behavior does not change and the uncertain output carries the same operational consequences as a confident one. Because agents lack mechanisms to detect low-confidence reasoning, they do not pause execution to request clarification, reconsider assumptions, expand the search space, or branch the workflow. As a result, ambiguous or underspecified steps often trigger confident hallucinations that steer the entire pipeline in an incorrect direction, with errors compounding across subsequent stages. Future research may therefore focus on developing uncertainty-aware workflow planning, where agents explicitly model their confidence and adapt their behavior accordingly. Such systems could flag uncertain decisions, trigger query refinement, invoke stronger verification modes, or explore multiple candidate workflows when ambiguity is high. Introducing uncertainty as a first-class signal would allow agents to dynamically modulate autonomy and mitigate the error-propagation patterns that are pervasive in long-horizon data science tasks.
> >
> > **6.3 Pipeline-Level Reflection**
> >
> > A further limitation is that existing reflection mechanisms operate only at the step level. Reviewers check the latest code snippet, self-reflection revises the previous response, and unit tests validate single functions. Yet data science workflows are inherently multi-stage, and small upstream issues, such as incorrect filtering, misaligned merges, omitted preprocessing, often propagate silently into feature engineering, model training, and evaluation. Current agents lack the ability to revisit earlier stages, attribute downstream errors to upstream decisions, or assess the global coherence of the entire analytical pipeline. Future research may address this gap by developing pipeline-level reflection: constructing explicit workflow graphs, tracing how information and errors flow across steps, identifying high-risk nodes, performing stage-level sanity checks, and enabling global revisions that correct the pipeline holistically rather than locally. Such mechanisms would transform reflection from shallow post-hoc editing into a principled, system-level reliability process.

---

### Review · Reviewer_Rh6q · 2025-11-03

**Summary Of Contributions:**

This survey paper provides a comprehensive analysis of Large Language Model (LLM)-based agents designed for data science tasks. The authors adopt a dual-perspective framework:
- (1) from the agent design perspective, examining agent roles, execution structures, external knowledge integration, and reflection mechanisms; and
- (2) from the data science perspective, analyzing how agents are applied across key workflow stages including data preprocessing, model development, evaluation, and visualization.
The paper also provides an extensive review of benchmarks and discusses future research opportunities.

**Audience:**

Yes

**Audience Explanation:**

1. While, this paper indeed presents a lot of agent works with comprehensive taxonimies, and the contents can be useful for the beginners to learn the context of existing research.
2.Table 4 provides an extensive compilation of 35+ benchmarks with details on their sources, number of tasks, and task types, which is highly valuable for researchers.
3. The survey covers a broad range of recent work with detailed tables summarizing different frameworks and their characteristics (e.g., Tables 1-4), making it a valuable reference resource.

**Claims And Evidence:**

Yes

**Claims Explanation:**

1. The paper's organization around both agent design principles and data science workflows is well-structured and provides a holistic view of the field. This dual lens effectively bridges general agent architectures with domain-specific applications.
2. The paper provides detailed categorizations across multiple dimensions: Agent roles (single, two-agent, multi-agent, dynamic), Execution structures (static vs. dynamic), External knowledge methods (databases, retrieval, APIs), Reflection mechanisms (agent feedback, code error handling, unit testing, etc.)

**Requested Changes:**

1. As a survey, this paper mainly listed some existing works and organize them well. However, it seems that there is no quantative analysis tha help people understand better for the overall trending, such as keywork anlaysis, benchmark performnace thrending vs dates, etc.
2. Please add more insights beyond a simple categorization. For example, what are the effective methods that able to be scaled? What's the limitation of the method? Are these methods only applicable to open-source models or close-source models (i.e. will better model backbone bering better performance).
3. Section 6 on "Future Research Opportunity" is underdeveloped (only 3 subsections covering ~30 lines). Given the comprehensive survey of existing work, more substantive guidance on open problems and research gaps would be valuable.

---

> ### Author Response · Authors · 2025-11-23
> **Response to requested change 1 (Lack of quantitative analysis)**
>
> We fully agree that the original version lacked quantitative analyses that could help readers better understand the broader landscape. In the revised manuscript, we have incorporated two forms of quantitative visualization to address this concern:
>
> **(1) Keyword frequency cloud (added at the beginning of Section 3).** We selected 61 representative papers from the corpus we reviewed, and extracted the frequencies of core technical keywords such as *multi-agent*, *MCTS*, *RAG*, *domain knowledge*, etc. Each keyword is colored according to its corresponding subsection in Sections 3.1–3.4, allowing readers to visually grasp the relative importance and prevalence of different methodological components. While figures cannot be included in the response, we provide the underlying data used to generate the cloud visualization in Table 1 below.
>
> **(2) Temporal SOTA performance trends on benchmarks (added in Section 5).** To illustrate benchmark evolution, we included line charts depicting SOTA performance over time for three representative benchmarks—InfiAgent-DABench, DSBench, and ML-Bench. Each data point is annotated with the corresponding system and backbone LLM. These visualizations help readers intuitively observe how performance has progressed over time and identify the current leading systems for each benchmark. In table 2, we include the statistics used for the InfiAgent-DABench plot as an example.
>
>
> **Table 1**
>
> | AgentRole Key | Freq | ExecStruct Key | Freq | ExternalKnow Key | Freq | Reflection Key | Freq |
> | --- | --- | --- | --- | --- | --- | --- | --- |
> | multi-agent | 397 | react | 236 | rag | 1486 | reflection | 139 |
> | coder | 229 | mcts | 55 | retrieval | 313 | debugging | 123 |
> | planner | 101 | task graph | 28 | domain knowledge | 105 | checkpoint | 90 |
> | developer | 79 | monte carlo tree search | 16 | case-based reasoning | 63 | unit tests | 47 |
> | executor | 66 | orchestration | 15 | metadata | 61 | test cases | 43 |
> | single agent | 56 | human-in-the-loop | 12 | knowledge base | 53 | human feedback | 31 |
> | reviewer | 52 | error handling | 10 | retrieval augmented generation | 40 | performance metrics | 28 |
> | task decomposition | 20 | fixed pipeline | 3 | retriever | 35 | self-reflection | 23 |
> | tester | 3 | hierarchical planning | 2 | web search | 31 | program repair | 3 |
> | qa engineer | 2 | graph-based planning | 2 | function calling | 25 | critic agent | 2 |
> | two-agent | 1 | rollback | 2 | api calls | 22 | history window | 2 |
> | task allocation | 1 | task routing | 1 | search engine | 19 | exception handling | 1 |
> | dynamic agents | 0 | static workflow | 0 | knowledge graph | 6 | agent feedback | 0 |
> | hierarchical agents | 0 | scripted pipeline | 0 | llamaindex | 6 | self-critique | 0 |
> | client-server agents | 0 | dynamic execution | 0 | tool calling | 6 | reviewer feedback | 0 |
> | controller agent | 0 | just-in-time planning | 0 | external database | 2 | regression tests | 0 |
> | manager agent | 0 | plan-then-execute | 0 | external memory | 1 | error diagnosis | 0 |
> | planner-executor | 0 | controller routing | 0 | experiment logs | 0 | metrics feedback | 0 |
> | minimum function agent | 0 | role-based routing | 0 | vector store | 0 | score-based optimization | 0 |
> | role specialization | 0 | error recovery | 0 | embedding index | 0 | experience replay | 0 |
>
>
> **Table 2**
> | Date | Performance | System | Backbone |
> | --- | --- | --- | --- |
> | 2024 Jan | 0.79 | Initial leaderboard | GPT-4 |
> | 2024 June | 0.88 | TaskWeaver | GPT-4 |
> | 2025 July | 0.949 | Data Interpreter | GPT-4o |
> | 2025 Oct | 0.953 | PharmaData-Agent | GPT-4.1 |

---

> > ### Author Response · Authors · 2025-11-23
> > **Revision 1 to requested change 2 (Need deeper analysis of methods and limitations)**
> >
> > We thank the reviewer for highlighting the need to provide deeper insights on the methods comparisons and limitations. In response, we have made substantial revisions to Section 3 and Section 5 to explicitly analyze the advantages, limitations, scalability, and applicability of different system-design methods.
> >
> > ## **Revision 1: Rewriting the introduction of Section 3.**
> >
> > We fully rewrote the opening paragraph of Section 3. Instead of using a generic description, we now motivate the entire section by discussing core limitations of current LLMs (e.g., hallucination, fragile code generation, and context forgetting), and explains why the design dimensions covered in Sections 3.1–3.4 are effective ways to mitigate these issues. This provides clearer motivation for our organization and creates a more natural lead-in to the methods.
> >
> > Revised text:
> >
> > > LLMs provide strong linguistic and generative capabilities, but they exhibit structural limitations that become particularly evident in data science workflows. First, LLMs are prone to hallucination, e.g., fabricating column names, statistical results, or entire processing steps. Second, LLM-generated code is brittle. Even when logically correct in natural language, produced scripts frequently break due to issues such as type inconsistencies, missing dependencies, unstable pipelines, or non-reproducible execution paths. Third, LLMs struggle with long-horizon tasks. Their limited ability to retain and manipulate extended context leads to goal drift, forgotten assumptions, and inconsistencies across multi-step analytical processes. These limitations show that raw LLMs are not reliable as end-to-end data science systems and need structured agent architectures to provide the capabilities they lack and to address common failure modes.\\
> > Motivated by these limitations, recent work has increasingly turned to agent architectures that introduce structure, modularity, and external support, enabling LLMs to compensate for their inherent limitations. For example, grounding and verification mechanisms help reduce hallucination, controlled execution improves reliability, and explicit role decomposition mitigates long-context instability. Agent systems thus do not merely wrap LLMs—they provide the organizational scaffolding needed to transform a general-purpose model into a dependable data science practitioner.\\
> > Seen from this perspective, our analysis focuses on four design dimensions: agent role design (3.1), execution structure (3.2), external knowledge integration (3.3), and reflection mechanisms (3.4). They form a structured solution to the inherent limitations of LLMs in data science settings. The remainder of this section examines each dimension in turn, showing how existing systems operationalize these design principles in practice.

---

> > > ### Author Response · Authors · 2025-11-23
> > > **Revision 2 to requested change 2 (Need deeper analysis of methods and limitations)**
> > >
> > > ## **Revision 2: Adding advantages, limitations, and trade-offs for each method.**
> > >
> > > For every method discussed in Section 3, we added an ending paragraph describing its strengths, weaknesses, and trade-offs. In addition, each subsection now concludes with a one-hot comparison table summarizing the method’s capabilities, limitations, and suitable application scenarios.
> > >
> > > Take **Section 3.1 Agent Role Design** as an illustrative example:
> > >
> > > - In *3.1.1 Single Agent*, we added an ending paragraph:
> > > > Single-agent designs offer minimal overhead and are easy to deploy, making them attractive for low-latency or low-cost settings. However, since all reasoning, memory, and execution are concentrated in a single model, they inherit most of the LLM limitations: hallucinations are unmitigated, generated code is fragile, and long-horizon tasks often suffer from goal drift and lost context. As a result, single-agent systems are often insufficient for complex or multi-stage workflows that require long-horizon reasoning, consistent state tracking, or robust code execution.
> > > - In *3.1.2 Two Agents*, the ending paragraph is:
> > > > Two-agent designs improve over single-agent systems by splitting responsibilities into complementary roles. Planner–executor structures provide more structured task decomposition, and coder–reviewer structures introduce lightweight verification, catching some errors or hallucinations that a single agent would miss. However, the approach remains sensitive to failures such as a planner hallucinating an incorrect plan or a reviewer failing to identify subtle logical errors. Communication cost also increases, and the reliability gains depend heavily on how well the two roles align. As a result, two-agent systems offer a moderate balance between simplicity and robustness, but their effectiveness is bounded by the underlying LLM's consistency and self-correction ability.
> > > - For *3.1.3 Multiple agents* we added:
> > > > Multi-agent architectures offer the highest scalability and strongest resistance to long-context forgetting, as each agent operates within a narrowly scoped role.
> > > They can achieve high reliability through division of labor, modular workflows, and cross-agent verification.
> > > Yet the increased number of agents introduces coordination overhead, susceptibility to miscommunication, and longer execution latency.
> > > These systems are powerful but harder to control, and their effectiveness depends on careful manual workflow design.
> > > - For *3.1.4 Dynamic Agents* we added the ending paragraph:
> > > > Dynamic agent generation offers high flexibility by creating agents on demand rather than relying on manually designed roles, prompts, and pipeline. This allows the system to adjust its structure at runtime and better accommodate diverse or evolving data science tasks. However, dynamically spawning agents also amplifies LLM uncertainty, reduces reproducibility, and increases computational and coordination overhead. As a result, these systems provide strong adaptability but exhibit lower predictability and stability, making them challenging to deploy in stable production settings.
> > > - In *3.1.5 Summary of Agent Role*, we included a cross-method comparison table:
> > >
> > >
> > > ### Trade-off summary across agent role designs
> > > **Legend:** ⬤ = strong, ◑ = moderate, ◯ = weak
> > >
> > > | **Dimension** | **Single Agent** | **Two-Agent** | **Multi-Agent** | **Dynamic Agents** |
> > > |---------------|------------------|----------------|------------------|----------------------|
> > > | **Reliability**          | ◯ | ◑ | ⬤ | ◑ |
> > > | **Scalability**          | ◯ | ◑ | ⬤ | ⬤ |
> > > | **Coordination Cost**    | ⬤ | ◑ | ◯ | ◯ |
> > > | **Predictability/Stabillity**       | ◑ | ◑ | ⬤ | ◯ |
> > > | **Industrial Applicability** | Quick analyses, ad-hoc queries, and low-risk automation | Medium-scale workflows requiring basic verification and predictable execution | Production pipelines with modular stages, quality checks, and stable data dependencies | Exploratory analytics, quick pipeline prototyping, or changeable environment |

---

> > > > ### Author Response · Authors · 2025-11-23
> > > > **Revision 3 to requested change 2 (Need deeper analysis of methods and limitations)**
> > > >
> > > > ## **Revision 3: Expanding Section 5 on benchmark limitations.**
> > > >
> > > > We also revised Section 5 to provide a more analytical discussion of the limitations in current benchmark evaluations. The updated text identifies three common issues that may cause benchmark results to mischaracterize an agent’s true capabilities:
> > > >
> > > > 1. evaluating only final outputs without accounting for error accumulation across intermediate steps,
> > > > 2. rigid task formats that encourage pattern learning rather than genuine data understanding and reasoning,
> > > > 3. problem settings that are overly clean and do not reflect realistic complexity.
> > > >
> > > > Revised text:
> > > >
> > > > > A wide range of benchmarks has been developed to evaluate LLM-based data science agents, each reflecting different assumptions about what “data science competence” entails. Table 5 summarizes the major suites. General-purpose benchmarks such as ML-Bench and DSBench assess core analytical skills across preprocessing, modeling, feature engineering, and multimodal reasoning. Other benchmarks specialize in particular workflow stages, such as visualization in MatPlotAgent-Bench or geospatial analysis in GeoAgent-Bench. Domain-focused benchmarks, such as FoodPuzzle, GenoTEX, or AgentClinic, measure scientific and data-driven reasoning under discipline-specific constraints. In addition, TheAgentCompany provides a practical evaluation framework covering end-to-end automation tasks in corporate environments, including data science, engineering, and business operations.
> > > >
> > > > > While these benchmarks provide broad coverage and standardized evaluation settings, they also exhibit structural limitations that may mischaracterize an agent’s true capabilities. First, most datasets evaluate short, isolated tasks. Suites such as DSBench or ML-Bench typically test a single operation (e.g., computing a statistic, performing a small transformation, or fitting a lightweight model), thus missing the errors accumulated across multi-stage workflows. Second, rigid task formats allow models to exploit surface regularities rather than reasoning. In MatPlotAgent-Bench, for example, many tasks share highly similar visualization templates (e.g., generating a 2×3 grid of boxplots), so an agent may match the template instead of demonstrating genuine analytical understanding and reasoning. Third, benchmark inputs are often clean, complete, and unambiguous. But LLM agents frequently fail when inputs are messy or underspecified (e.g., irregular schemas or missing values), a scenario that current benchmarks like GeoAgent-Bench still rarely capture.
> > > >
> > > > > Each benchmark therefore captures only a particular slice of the capabilities required for real data science practice. General-purpose suites test broad coverage but not long-horizon robustness; workflow-specific and domain-specific benchmarks evaluate deeper skills but in narrower settings. A comprehensive evaluation therefore requires viewing performance across multiple benchmarks, each revealing different aspects of reliability, reasoning, and robustness.

---

> > > > > ### Author Response · Authors · 2025-11-23
> > > > > **Response to requested change 3 (Underdeveloped future works)**
> > > > >
> > > > > Given the breadth of the survey, we agree that the Future Research Opportunity section should offer a more substantive and analytically grounded discussion of research gaps. To address this, we substantially expanded it, enriched its conceptual depth, and ensured that each proposed direction is explicitly motivated by the limitations synthesized in the preceding sections.
> > > > >
> > > > > In the revision, we removed two subsections. The earlier subsection on **Multimodal Processing** did not correspond to any major limitation discussed earlier, and the idea of **Trainable Architecture** has since been explored by recent work published after our initial drafting of this survey. We therefore eliminated both topics.
> > > > >
> > > > > The previous subsection on **Advanced Reflection Mechanisms** was also overly broad. Our original intention was to address failures in long-horizon reasoning and error propagation throughout multi-step workflows. To make it more precise, we reframed this direction as **Pipeline-Level Reflection**, which more accurately describes the long-horizon error localization and correction that current systems lack.
> > > > >
> > > > > In addition to refining and narrowing our previous content, we also introduced two research directions that directly reflect the limitations identified throughout the survey and are not addressed by existing work. The first is *Data-Centric Diagnostics*, which targets the data-side failure modes highlighted in Sections 3 and 5, such as schema irregularities, type mismatches, and distributional anomalies that current reflection mechanisms cannot detect. The second is *Uncertainty-Aware Workflow Planning*, motivated by our observation that present-day agents execute entire pipelines without any awareness of confidence or ambiguity, resulting in error propagation when intermediate reasoning steps are unreliable. These two directions respond directly to gaps we identified in current systems and offer practical, unexplored avenues for improving the robustness of LLM-based data science agents.
> > > > >
> > > > > The revised text is as follows:
> > > > >
> > > > > > **6.1 Data-Centric Diagnostics**
> > > > >
> > > > > >As discussed earlier, many high-impact failures in LLM-based data science agents originate not from the agent's reasoning, but from properties of the data itself. Agents frequently misinterpret ... (examples and explainations). Current reflection mechanisms are blind to such data issues... This limitation explains why agents often produce ``correctly reasoned'' but fundamentally invalid analytical results. Future research should therefore develop data-centric diagnostic modules that automatically evaluate schema consistency, detect anomalous or distribution-shifted subsets, identify type or semantic mismatches, and trace how specific data defects influence downstream modeling outcomes. By elevating data quality analysis to a first-class reflective capability, agents would be better equipped to prevent systematic failure modes that current systems cannot even observe.
> > > > >
> > > > > > **6.2 Uncertainty-Aware Workflow Planning**
> > > > >
> > > > > > Another highlighted limitation is that current LLM-based agents operate without any awareness of uncertainty. When the model is unsure about ... (examples), its downstream behavior does not change and the uncertain output carries the same operational consequences as a confident one. Because agents lack ...(disadvantages of current methods). As a result, ambiguous or underspecified steps often trigger confident hallucinations that steer the entire pipeline in an incorrect direction, with errors compounding across subsequent stages. Future research may therefore focus on developing uncertainty-aware workflow planning, where agents explicitly model their confidence and adapt their behavior accordingly. Such systems could flag uncertain decisions, trigger query refinement, invoke stronger verification modes, or explore multiple candidate workflows when ambiguity is high. Introducing uncertainty as a first-class signal would allow agents to dynamically modulate autonomy and mitigate the error-propagation patterns that are pervasive in long-horizon data science tasks.
> > > > >
> > > > > >**6.3 Pipeline-Level Reflection**
> > > > >
> > > > > >A further limitation is that existing reflection mechanisms operate only at the step level. Reviewers check ...(examples). Yet data science workflows are inherently multi-stage, and small upstream issues, such as ...(examples). Current agents lack the ability to revisit earlier stages, attribute downstream errors to upstream decisions, or assess the global coherence of the entire analytical pipeline. Future research may address this gap by developing pipeline-level reflection: constructing explicit workflow graphs, tracing how information and errors flow across steps, identifying high-risk nodes, performing stage-level sanity checks, and enabling global revisions that correct the pipeline holistically rather than locally. Such mechanisms would transform reflection from shallow post-hoc editing into a principled, system-level reliability process.

---

### Decision · Action_Editor_sbYx · 2026-01-04

**Recommendation:** Accept with minor revision

**Additional Comments:**

I recommend accepting this manuscript with minor revisions to ensure the revisions presented in the author response comments are properly integrated into the manuscript (e.g. all "revision" headings).

**Audience:**

Yes

**Audience Explanation:**

As all reviewers indicate, there will likely be an audience for this work in newcomers to the area who will benefit from the extensive categorization and works described / collected here.

**Claims And Evidence:**

Yes

**Claims Explanation:**

The manuscript delivers on its key claims -- presenting a framework for categorizing recent LLM-for-data science works. Reviewers disagree on what depth of analysis is necessary to meet the evidence requirement; however, the authors have provided extensive revision's described in the response comments to address these concerns.